# Demographic and Clinical Factors Associated With SARS-CoV-2 Spike 1 Antibody Response Among Vaccinated US Adults: the C4R Study

John S. Kim [1,2], Yifei Sun[3], Pallavi Balte[2], Mary Cushman [4,5], Rebekah Boyle[5], Russell P. Tracy[5], Linda M. Styer[6], Taison D. Bell[1], Michaela R. Anderson[7], Norrina B. Allen [8], Pamela J. Schreiner[9], Russell P. Bowler[10], David A. Schwartz[11], Joyce S. Lee[11], Vanessa Xanthakis[12,13], Margaret F. Doyle[5], Elizabeth A. Regan[14], Barry J. Make[10], Alka M. Kanaya[15], Sally E. Wenzel [16], Josef Coresh [17,18], Carmen R. Isasi [19], Laura M. Raffield [20], Mitchell S. V. Elkind[21,22], Virginia J. Howard[23], Victor E. Ortega[24], Prescott Woodruff[25], Shelley A. Cole[26], Joel M. Henderson[27], Nicholas J. Mantis [6,28], Monica M. Parker[6], Ryan T. Demmer[9,22,29,30] ✉ & Elizabeth C. Oelsner [2,30] ✉

This study investigates correlates of anti-S1 antibody response following COVID-19 vaccination in a U.S. population-based meta-cohort of adults participating in longstanding NIH-funded cohort studies. Anti-S1 antibodies were measured from dried blood spots collected between February 2021-August 2022 using Luminex-based microsphere immunoassays. Of 6245 participants, mean age was 73 years (range, 21-100), 58% were female, and 76% were non-Hispanic White. Nearly 52% of participants received the BNT162b2 vaccine and 48% received the mRNA-1273 vaccine. Lower anti-S1 antibody levels are associated with age of 65 years or older, male sex, higher body mass index, smoking, diabetes, COPD and receipt of BNT16b2 vaccine (vs mRNA-1273). Participants with a prior infection, particularly those with a history of hospitalized illness, have higher anti-S1 antibody levels. These results suggest that adults with certain socio-demographic and clinical characteristics may have less robust antibody responses to COVID-19 vaccination and could be prioritized for more frequent re-vaccination.

COVID-19 was the third leading cause of death in the U.S. in 2020 and 2021[1]. COVID-19 vaccination is an important public health intervention to prevent infection and attenuate illness severity[2,3]. The primary mechanism of messenger RNA (mRNA) vaccines, BNT162b2 (Pfizer-BioNTech) and mRNA-1273 (Moderna), is the production of antibodies against the spike protein of SARS-CoV-2[4]. Some studies have shown differential antibody responses to vaccination and waning antibody levels over time[5–7]. Variation in vaccine-induced antibody response may be clinically significant since lower humoral responses to SARS-CoV-2 vaccines are predictive of higher risk for breakthrough infections and severe COVID-19 clinical outcomes[6,8–10]. Hence, identification of individuals at risk of lower vaccine-induced antibody response and faster antibody waning could inform personalized vaccination strategies.

Most prior studies on vaccine immunogenicity leveraged clinical trials or highly selected cohorts (e.g., health-care workers and residents of long-term care facilities)[6,11–13]. A general-population study conducted in the United Kingdom (UK), where the majority of individuals received the AstraZeneca vaccine, observed lower rates of post-vaccination seropositivity in older adults, males, and those with chronic health conditions[5,14,15]. However, the determinants of post-vaccination antibody response, including magnitude and durability of antibody levels, have not been comprehensively investigated in multi-ethnic, diverse, population-based U.S. samples where mRNA vaccines predominate.

This study aimed to identify correlates of anti-S1 IgG antibody levels after COVID-19 vaccination with mRNA vaccines in the Collaborative Cohort of Cohorts for COVID-19 Research (C4R)[16]. C4R is a national prospective study of U.S. adults participating in 14 longitudinal cohort studies that collectively constitute a large, well-characterized, population-based sample. Anti-S1 IgG antibody levels were measured by serosurvey and examined with respect to pre-pandemic and pandemic-era socio-demographic and clinical factors. Correlates of antibody levels over the period of time since vaccination were elucidated.

## Results
### Participant characteristics
There were 6245 participants who received two doses of a mRNA COVID-19 vaccine with measured anti-S1 IgG antibody levels (Table 1, Supplementary Fig. 1). The mean time between the first vaccine dose and serosurvey was 4.0 months (SD 1.7; range 0.1–7.1). Mean age was 73 years (range, 21–100), 76.9% of participants were aged ≥65 years, 58.3% were female, 76.4% self-identified as non-Hispanic White, 17.6% as African American/Black, 3.1% as Asian, 2.0% as American Indian, and 0.9% as Hispanic/Latino. 51.8% of participants received the BNT162b2 vaccine and 48.2% received the mRNA-1273 vaccine. Infection prior to serosurvey was self-reported in 18% of C4R participants and 4.8% self-reported infection after vaccination but before serosurvey participation. Of note, compared to the participants eligible for this report, C4R participants who self-reported vaccination but did not complete the serosurvey were younger, more racially and ethnically diverse, and less likely to have received the mRNA-1273 vaccine (Supplementary Table 1).

### Antibody responses
Among the samples, 97% showed antibody reactivity above the laboratory threshold used to confirm prior SARS-CoV-2 exposure (Supplementary Table 2). Nonetheless, there was a large range of continuous antibody levels across the sample (Table 1). Participants in the highest quartile of anti-S1 levels were more likely to be female, younger, recipients of the mRNA-1273 vaccine, and report prior infection (Table 1). Antibody levels were highest at 60 days following the first vaccine dose, after which antibody levels waned, on average, at a rate of 21% per month (Fig. 1). Of note, based on current vaccine recommendations, 60 days following the first vaccine dose corresponds to 32 (BNT162b2) to 39 (mRNA-1273) days following the second dose.

### Correlates of antibody response
Both unadjusted and multivariable-adjusted models consistently demonstrated associations between antibody levels and several participant characteristics (Table 2). For correlates that were significantly associated with antibody levels in multivariable-adjusted models, associations with antibody levels were plotted over time since the first vaccine dose using linear splines (Figs. 2–4).

Lower antibody levels were observed in older participants, men, and participants with a history of obesity, smoking, diabetes, or COPD (Fig. 2, Table 2). AIAN and Asian race were associated with a lower and greater antibody response, respectively, compared to Non-Hispanic White race. However, the confidence intervals were wide in the setting of relatively small sample sizes for these groups. In models testing for differences in the slope of anti-S1 IgG waning, the rate of antibody decline increased monotonically in smokers. Current smokers realized an anti-S1 IgG decline that was 6.8% per month (95% CI, 1.0–12.1) greater than never smokers, adjusting for other covariates. Other socio-demographic and pre-pandemic clinical factors were not consistently associated with rate of antibody waning (Supplementary Table 3).

With respect to vaccine history, receipt of the mRNA-1273 vaccine was associated with a higher antibody response compared to receipt of the BNT162b2 vaccine (Fig. 3). The BNT162b2 vaccine was also associated with faster antibody waning versus the mRNA-1273 vaccine (Supplementary Table 3).

Compared to uninfected participants, those with a self-reported history of infection prior to vaccination showed higher antibody levels, particularly those with a history of hospitalization with COVID-19 (Fig. 4). Anti-nucleocapsid antibody level, an objective marker of natural infection, was also associated with higher anti-S1 IgG. Infections occurring after vaccination (sometimes called "breakthrough" infections), but prior to the serosurvey, were not associated with antibody levels after multivariable adjustment (Fig. 4, Table 2). Infections after vaccination were associated with a lesser rate of antibody waning, as was higher anti-nucleocapsid antibody titer (Supplementary Table 3), which could be due to the fact that time since vaccination does not correspond to time since most recent SARS-CoV-2 antigen exposure in this group.

### Sensitivity analyses
Results were similar in complete case analyses (Supplementary Table 4) and in models excluding cases of self-reported infection that lacked confirmatory testing (Supplementary Table 5). Findings were also consistent in models using different approaches to assess potential cohort heterogeneity, including models adjusting for cohort (Supplementary Table 6) and stratified models that were restricted to, or excluded, the three cohorts (ARIC, FHS, REGARDS) contributing the largest number of participants to the main analysis (Supplementary Fig. 2a–d).

## Discussion
Antibody responses to COVID-19 vaccination varied according to socio-demographic and clinical characteristics in a large, multi-ethnic, US population-based cohort, in which all participants received mRNA-based vaccines. Male sex, obesity, smoking history, diabetes, and COPD were associated with significantly lower antibody responses. Moreover, even after accounting for clinical conditions, smoking was associated with both lower peak antibody responses to vaccination and faster antibody waning. A history of SARS-CoV-2 infection and receipt of the mRNA-1273 vaccine were associated with higher antibody responses. Our findings suggest clinical characteristics that could be used to inform recommendations regarding re-vaccination.

Compared to prior studies, many of which included individuals with limited information on pre-pandemic health conditions[9,14,15,17], our work is distinguished by its use of prospectively-collected pre-pandemic and pandemic-era risk factor data in racially and ethnically diverse U.S. cohorts. Our study design reduces the potential role of confounding by unmeasured pre-pandemic phenotypes and enhances generalizability to the U.S. population. C4R findings also extend knowledge regarding real-world population responses to mRNA vaccines. UK population-based studies were comprised mostly of individuals who received the BNT162b2 or ChAdOx1 vaccines; in contrast, our study sample included participants who received mRNA vaccines, and approximately half received the mRNA-1273 vaccine[15,17]. Compared to BNT162b2, the mRNA-1273 vaccine was associated with substantially higher antibody responses and slower antibody waning. Whether this

**Table 1 | Baseline characteristics of 6245 C4R participants with serology results from February 2021 through August 2022**

| Characteristic | Overall | Anti-S1 quartiles | | | |
|---|---|---|---|---|---|
| | | Quartile 1 | Quartile 2 | Quartile 3 | Quartile 4 |
| No. of participants | 6245 | 1562 | 1561 | 1561 | 1561 |
| Anti-S1 antibody, mean (SD), MFI (log-transformed) | 8.4 (1.3) | 6.6 (1.0) | 8.1 (0.3) | 9.0 (0.2) | 9.9 (0.3) |
| Natural log transformed-anti-N antibody, mean (SD), MFI | 4.7 (1.1) | 4.3 (0.8) | 4.5 (0.9) | 4.7 (1.0) | 5.3 (1.5) |
| Age, no. (%) | | | | | |
| Less than 65 years | 1448 (23.2%) | 235 (15%) | 374 (24%) | 430 (27.5%) | 409 (26.2%) |
| 65–79 years | 3182 (51%) | 868 (55.6%) | 846 (54.2%) | 747 (47.9%) | 721 (46.2%) |
| 80 years and greater | 1615 (25.9%) | 459 (29.4%) | 341 (21.8%) | 384 (24.6%) | 431 (27.6%) |
| Female sex, no. (%) | 3643 (58.3%) | 784 (50.2%) | 906 (58%) | 964 (61.8%) | 989 (63.4%) |
| Self-reported race or ethnicity, no. (%) | | | | | |
| Non-Hispanic White | 4773 (76.4%) | 1219 (78%) | 1172 (75.1%) | 1196 (76.6%) | 1186 (76%) |
| American Indian and Alaskan Native | 125 (2%) | 30 (1.9%) | 22 (1.4%) | 26 (1.7%) | 47 (3%) |
| Asian | 193 (3.1%) | 28 (1.8%) | 51 (3.3%) | 66 (4.2%) | 48 (3.1%) |
| African American/Black | 1098 (17.6%) | 271 (17.3%) | 302 (19.3%) | 258 (16.5%) | 267 (17.1%) |
| Hispanic | 56 (0.9%) | 14 (0.9%) | 14 (0.9%) | 15 (1%) | 13 (0.8%) |
| Education attainment, no. (%) | | | | | |
| Less than high school | 245 (3.9%) | 52 (3.3%) | 60 (3.8%) | 57 (3.7%) | 76 (4.9%) |
| High school | 1358 (21.7%) | 336 (21.5%) | 303 (19.4%) | 329 (21.1%) | 390 (25%) |
| College | 1218 (19.5%) | 328 (21%) | 359 (23%) | 299 (19.2%) | 232 (14.9%) |
| Beyond college | 3424 (54.8%) | 846 (54.2%) | 839 (53.7%) | 876 (56.1%) | 863 (55.3%) |
| Study cohort, no. (%) | | | | | |
| ARIC | 1705 (27.3%) | 253 (16.2%) | 286 (18.3%) | 491 (31.5%) | 675 (43.2%) |
| CARDIA | 144 (2.3%) | 34 (2.2%) | 49 (3.1%) | 36 (2.3%) | 25 (1.6%) |
| COPDGene | 444 (7.1%) | 144 (9.2%) | 132 (8.5%) | 93 (6%) | 75 (4.8%) |
| FHS | 1138 (18.2%) | 127 (8.1%) | 258 (16.5%) | 326 (20.9%) | 427 (27.4%) |
| JHS | 34 (0.5%) | 7 (0.4%) | 6 (0.4%) | 14 (0.9%) | 7 (0.4%) |
| MASALA | 145 (2.3%) | 12 (0.8%) | 40 (2.6%) | 57 (3.7%) | 36 (2.3%) |
| MESA | 215 (3.4%) | 77 (4.9%) | 65 (4.2%) | 52 (3.3%) | 21 (1.3%) |
| PrePF | 72 (1.2%) | 24 (1.5%) | 18 (1.2%) | 16 (1%) | 14 (0.9%) |
| REGARDS | 2087 (33.4%) | 811 (51.9%) | 649 (41.6%) | 418 (26.8%) | 209 (13.4%) |
| SARP | 48 (0.8%) | 14 (0.9%) | 12 (0.8%) | 9 (0.6%) | 13 (0.8%) |
| SHS | 117 (1.9%) | 28 (1.8%) | 20 (1.3%) | 24 (1.5%) | 45 (2.9%) |
| SPIROMICS | 96 (1.5%) | 31 (2%) | 26 (1.7%) | 25 (1.6%) | 14 (0.9%) |
| Smoking status, no. (%) | | | | | |
| Never | 2839 (45.5%) | 634 (40.6%) | 709 (45.4%) | 766 (49.1%) | 730 (46.8%) |
| Former | 2828 (45.3%) | 764 (48.9%) | 692 (44.3%) | 667 (42.7%) | 705 (45.2%) |
| Current | 578 (9.2%) | 164 (10.5%) | 160 (10.3%) | 128 (8.2%) | 126 (8.0%) |
| Body mass index, kg/m$^2$, no. (%) | | | | | |
| <25 kg/m$^2$ | 1602 (25.7%) | 403 (25.8%) | 394 (25.2%) | 390 (25%) | 415 (26.6%) |
| 25–29.9 kg/m$^2$ | 2455 (39.3%) | 584 (37.4%) | 612 (39.2%) | 639 (40.9%) | 620 (39.7%) |
| 30–35 kg/m$^2$ | 1357 (21.7%) | 338 (21.6%) | 357 (22.9%) | 325 (20.8%) | 337 (21.6%) |
| >35 kg/m$^2$ | 831 (13.3%) | 237 (15.2%) | 198 (12.7%) | 207 (13.3%) | 189 (12.1%) |
| Hypertension, no. (%) | 3430 (54.9%) | 969 (62%) | 842 (53.9%) | 804 (51.5%) | 815 (52.2%) |
| Diabetes, no. (%) | 1202 (19.2%) | 383 (24.5%) | 288 (18.4%) | 256 (16.4%) | 275 (17.6%) |
| Cardiovascular disease, no. (%) | 700 (11.2%) | 223 (14.3%) | 188 (12%) | 149 (9.5%) | 140 (9%) |
| Chronic obstructive pulmonary disease, no. (%) | 602 (9.6%) | 198 (12.7%) | 157 (10.1%) | 134 (8.6%) | 113 (7.2%) |
| Asthma, no. (%) | 750 (12%) | 215 (13.8%) | 187 (12%) | 165 (10.6%) | 183 (11.7%) |
| Chronic kidney disease, no. (%) | 156 (2.5%) | 40 (2.6%) | 40 (2.6%) | 40 (2.6%) | 36 (2.3%) |
| COVID-19 infection prior to serosurvey, no. (%) | | | | | |
| No infection | 4823 (77.2%) | 1316 (84.3%) | 1236 (79.2%) | 1215 (77.8%) | 1056 (67.6%) |
| Infection prior to vaccine—not hospitalized | 910 (14.6%) | 155 (9.9%) | 230 (14.7%) | 238 (15.2%) | 287 (18.4%) |
| Infection prior to vaccine—hospitalized | 214 (3.4%) | 41 (2.6%) | 41 (2.6%) | 52 (3.3%) | 80 (5.1%) |
| Infection after vaccine—not hospitalized | 298 (4.8%) | 50 (3.2%) | 54 (3.5%) | 56 (3.6%) | 138 (8.8%) |

**Table 1 (continued) | Baseline characteristics of 6245 C4R participants with serology results from February 2021 through August 2022**

| Characteristic | Overall | Anti-S1 quartiles | | | |
|---|---|---|---|---|---|
| | | Quartile 1 | Quartile 2 | Quartile 3 | Quartile 4 |
| Vaccine type, no. (%) | | | | | |
| BNT162b2 mRNA (Pfizer-BioNTech) | 3237 (51.8%) | 1080 (69.1%) | 823 (52.7%) | 702 (45%) | 632 (40.5%) |
| mRNA-1273 (Moderna) | 3008 (48.2%) | 482 (30.9%) | 738 (47.3%) | 859 (55%) | 929 (59.5%) |
| Time between first vaccination dose and serosurvey collection, mean (SD), months | 4 (1.7) | 5 (1.5) | 4.6 (11.5) | 3.8 (1.6) | 2.8 (1.5) |

*ARIC* Atherosclerosis Risk in Communities Study, *CARDIA* Coronary Artery Risk Development in Young Adults, *COPDGene* Genetic Epidemiology of Chronic Obstructive Pulmonary Disease, *FHS* Framingham Heart Study, *JHS* Jackson Heart Study, *MASALA* Mediators of Atherosclerosis in South Asians Living in America, *MESA* Multi-Ethnic Study of Atherosclerosis, *MFI* median fluorescence intensity, *PrePF* Preclinical Pulmonary Fibrosis, *REGARDS* Reasons for Geographic and Racial Differences in Stroke, *SARP* Severe Asthma Research Program, *SHS* Strong Heart Study, *SPIROMICS* Subpopulations and Intermediate Outcomes in COPD Study.
Data presented are from multiple imputation and numbers are rounded to whole integer and percentages are rounded to nearest tenth decimal point.
Chronic kidney disease defined as estimated glomerular filtration rate below 45 mL/min/1.73m$^2$.

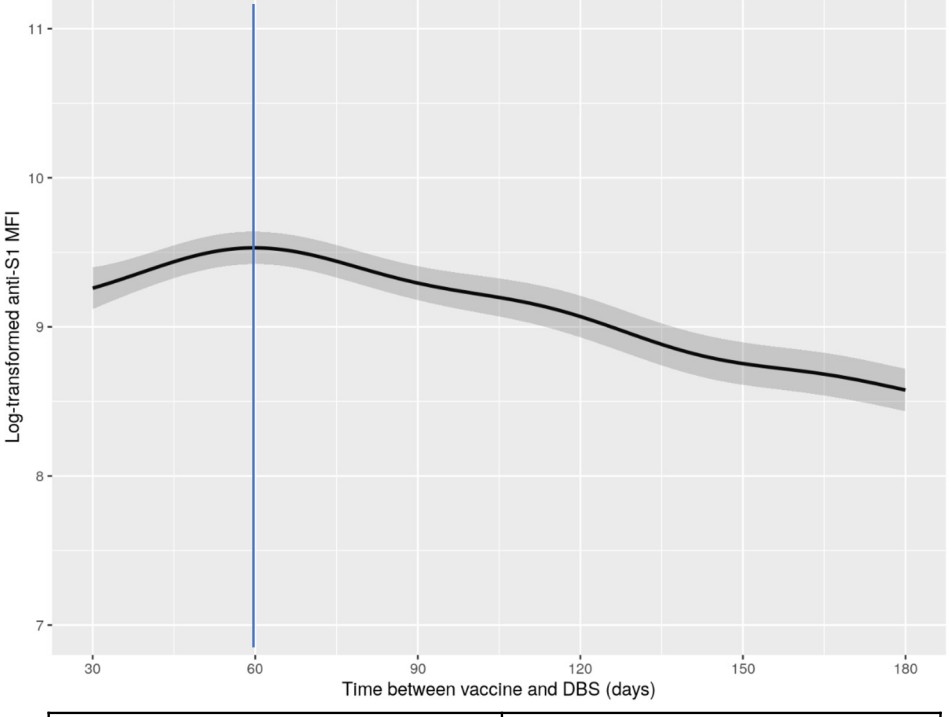

| Time from first vaccine dose to antibody measurement | Mean % change in anti-S1 IgG level rate of change per 30 days (95% CI) |
|---|---|
| Less than 60 days | 32.7 (16.4 to 51.3) |
| Greater than 60 days | -21.0 (-23.0 to -18.9) |

**Fig. 1 | Anti-S1 IgG levels by COVID-19 infection history over time between first dose of vaccine and antibody measurement.** The plot of predicted anti-S1 antibody levels generated from generalized additive model with smoothing splines adjusted for age, sex, race/ethnicity, smoking history, education attainment, body mass index, diabetes, cardiovascular disease, asthma, chronic obstructive pulmonary disease, anti-N, COVID-19 infection history, dried blood spot batch, and vaccine type. The vertical line indicates the 60-day mark. The X-axis was truncated at 30 days to ensure the inclusion of participants who had dried blood spot collection after receiving the second vaccine dose. Lighter color bands indicate 95% confidence intervals.

may be due to differences in dosing between the two mRNA vaccines for the primary series (mRNA-1273: 100 mcg/dose; BNT162b2: 30 mg/dose), differences in the recommended dosing schedule, or other differences in vaccine formulations merits further investigation. Our findings may also have implications for dosing considerations relating to additional "booster" vaccines, since Moderna decreased the "booster" dose to 50 mcg. Unfortunately, we did not have sufficient data to assess the antibody response to three or more doses of vaccine in this study; we intend to investigate this in the future using additional follow-up in C4R, which will yield repeated measures on vaccinations and antibody responses.

Relative to UK data, our C4R data show peak antibody levels occurring at approximately 60 days and antibody waning was evident by 90 days. This suggests that the U.S. peak was sooner than in the Virus Watch and REACT-2 cohorts, in which the peak occurred closer to 90 days after the first dose, but later than in the ONS cohort, in which the peak occurred approximately 28 days after the first dose[14,15,17]. Nonetheless, the timing of the peak antibody response in C4R was similar to that observed in a longitudinal U.S.-based study of BNT162b2 recipients[6]. In our U.S.-based cohort, waning occurred more rapidly relative to the U.K. population, in which anti-spike IgG levels showed no obvious sign of waning by 90 days. Of note, longer intervals between the first and second dose implemented in the U.K. (8–12 weeks) versus in the U.S. (3–4 weeks) have been hypothesized to provoke a more durable immune response; this issue deserves more careful attention in future research studies. The faster waning

**Table 2 | Correlates of anti-S1 IgG levels after COVID-19 vaccination in unadjusted and multivariable adjusted analyses**

| Correlates | Unadjusted | | Multivariable-adjusted | |
|---|---|---|---|---|
| | Estimated mean percent difference in anti-S1 IgG level (95% confidence interval) | P-value | Estimated mean percent difference in anti-S1 IgG level (95% confidence interval) | P-value |
| **Age** | | | | |
| Less than 65 years | 0.0 (ref) | | 0.0 (ref) | |
| 65–79 years | −28.9 (−34.5 to −22.7) | $6.32 \times 10^{-16}$ | −20 (−25.5 to −14) | $1.19 \times 10^{-9}$ |
| 80 years and greater | −28.2 (−34.6 to −21.1) | $6.13 \times 10^{-12}$ | −40.2 (−45.1 to −34.8) | $1.45 \times 10^{-31}$ |
| **Sex** | | | | |
| Female | 0.0 (ref) | | 0.0 (ref) | |
| Male | −23.9 (−28.8 to −18.6) | $1.39 \times 10^{-15}$ | −21.9 (−26.1 to −17.6) | $1.40 \times 10^{-18}$ |
| **Race/ethnicity** | | | | |
| Non-Hispanic white | 0.0 (ref) | | 0.0 (ref) | |
| American Indian or Alaskan Native | 22.1 (−3.5 to 54.8) | 0.10 | −26.2 (−40.3 to −8.8) | 0.005 |
| Asian | 30.0 (7.3 to 57.5) | 0.007 | 35 (15.6 to 57.8) | $1.57 \times 10^{-4}$ |
| Black | 1.3 (−7.1 to 10.6) | 0.77 | 4.4 (−3.1 to 12.4) | 0.26 |
| Hispanic | 2.6 (−27.7 to 45.8) | 0.88 | 10 (−17.1 to 45.8) | 0.51 |
| **Education attainment** | | | | |
| College or beyond | 0.0 (ref) | | 0.0 (ref) | |
| Less than high school | 12.9 (−5.1 to 34.2) | 0.17 | 0.5 (−12.7 to 15.7) | 0.94 |
| High school | 5.9 (−2.7 to 15.1) | 0.18 | −0.4 (−7 to 6.7) | 0.90 |
| Some college | −15.6 (−22.7 to −7.9) | $1.49 \times 10^{-4}$ | −1.4 (−8.1 to 5.9) | 0.70 |
| **Smoking history** | | | | |
| Never | 0.0 (ref) | | 0.0 (ref) | |
| Former | −12.3 (−18.1 to −5.9) | $2.27 \times 10^{-4}$ | −9.7 (−14.8 to −4.3) | $5.72 \times 10^{-4}$ |
| Current | −17.6 (−26.8 to −7.0) | 0.002 | −19.3 (−26.9 to −11.0) | $1.81 \times 10^{-5}$ |
| **Body mass index** | | | | |
| <25 kg/m$^2$ | 0.0 (ref) | | 0.0 (ref) | |
| 25–29.9 kg/m$^2$ | 3.4 (−5.1 to 12.6) | 0.44 | 6.8 (−0.3 to 14.6) | 0.06 |
| 30–34.9 kg/m$^2$ | −0.8 (−10.1 to 9.3) | 0.87 | 3.3 (−4.7 to 11.9) | 0.44 |
| >35 kg/m$^2$ | −10.3 (−19.9 to 0.4) | 0.06 | −12.0 (−20.0 to −3.1) | 0.009 |
| **Diabetes** | | | | |
| No | 0.0 (ref) | | 0.0 (ref) | |
| Yes | −21.0 (−27.4 to −14.1) | $3.56 \times 10^{-8}$ | −12.7 (−18.7 to −6.3) | $1.79 \times 10^{-4}$ |
| **Hypertension** | | | | |
| No | 0.0 (ref) | | 0.0 (ref) | |
| Yes | −17.1 (−22.4 to −11.3) | $3.78 \times 10^{-8}$ | −3.9 (−9.2 to 1.7) | 0.17 |
| **Cardiovascular disease** | | | | |
| No | 0.0 (ref) | | 0.0 (ref) | |
| Yes | −23.7 (−35.1 to −10.4) | 0.003 | −6.8 (−15.6 to 3.0) | 0.17 |
| **Chronic kidney disease** | | | | |
| No | 0.0 (ref) | | 0.0 (ref) | |
| Yes | −7.7 (−27.8 to 17.9) | 0.52 | −10.2 (−25.8 to 8.7) | 0.27 |
| **Asthma** | | | | |
| No | 0.0 (ref) | | 0.0 (ref) | |
| Yes | −12.2 (−24.0 to 1.4) | 0.08 | −7.6 (−16.5 to 2.2) | 0.13 |
| **Chronic obstructive pulmonary disease** | | | | |
| No | 0.0 (ref) | | 0.0 (ref) | |
| Yes | −29.4 (−38.9 to −18.5) | $1.49 \times 10^{-5}$ | −18.3 (−26.8 to −8.7) | $5.69 \times 10^{-4}$ |
| Log-transformed anti-N MFI (per 1-unit increment) | 42.2 (38.3 to 46.2) | $1.50 \times 10^{-129}$ | 37.4 (33.2 to 41.8) | $1.01 \times 10^{-64}$ |
| **COVID-19 infection prior to serosurveycollection** | | | | |
| No infection | 0.0 (ref) | | 0.0 (ref) | |
| Infection prior to vaccine-not hospitalized | 52.3 (38.0 to 68.0) | $4.40 \times 10^{-16}$ | 58.7 (44.5 to 74.2) | $2.42 \times 10^{-20}$ |
| Infection prior to vaccine-hospitalized | 67.0 (35.4 to 106.1) | $4.90 \times 10^{-6}$ | 80.9 (51.1 to 116.4) | $2.32 \times 10^{-9}$ |
| Infection after vaccine | 90.4 (57.6 to 130.0) | $3.40 \times 10^{-9}$ | −0.7 (−17.2 to 19.2) | 0.94 |

**Table 2 (continued) | Correlates of anti-S1 IgG levels after COVID-19 vaccination in unadjusted and multivariable adjusted analyses**

| Correlates | Unadjusted | | Multivariable-adjusted | |
| --- | --- | --- | --- | --- |
| | Estimated mean percent difference in anti-S1 IgG level (95% confidence interval) | *P*-value | Estimated mean percent difference in anti-S1 IgG level (95% confidence interval) | *P*-value |
| COVID-19 vaccine type | | | | |
| BNT162b2 mRNA (Pfizer-BioNTech) | 0.0 (ref) | | 0.0 (ref) | |
| mRNA-1273 (Moderna) | 77.9 (66.7 to 89.8) | 1.06e10$^{-66}$ | 90.4 (80.6 to 100.6) | 5.19e10$^{-123}$ |
| Time between vaccine and serosurvey collection (per 30 days) | −26.4 (−27.7 to −25.1) | 1.35e10$^{-239}$ | −17.7 (−19.7 to −15.8) | 3.40e10$^{-58}$ |

*MFI* median fluorescence intensity.
Chronic kidney disease defined as estimated glomerular filtration rate below 45 mL/min/1.73 m.
Multivariable adjusted results arise from a linear regression model including all variables present in this table.
Statistical significance was assessed using two-sided tests. *P*-values are reported without adjustment for multiple comparisons.
Results from 6245 C4R participants with serology results from February 2021 through August 2022.

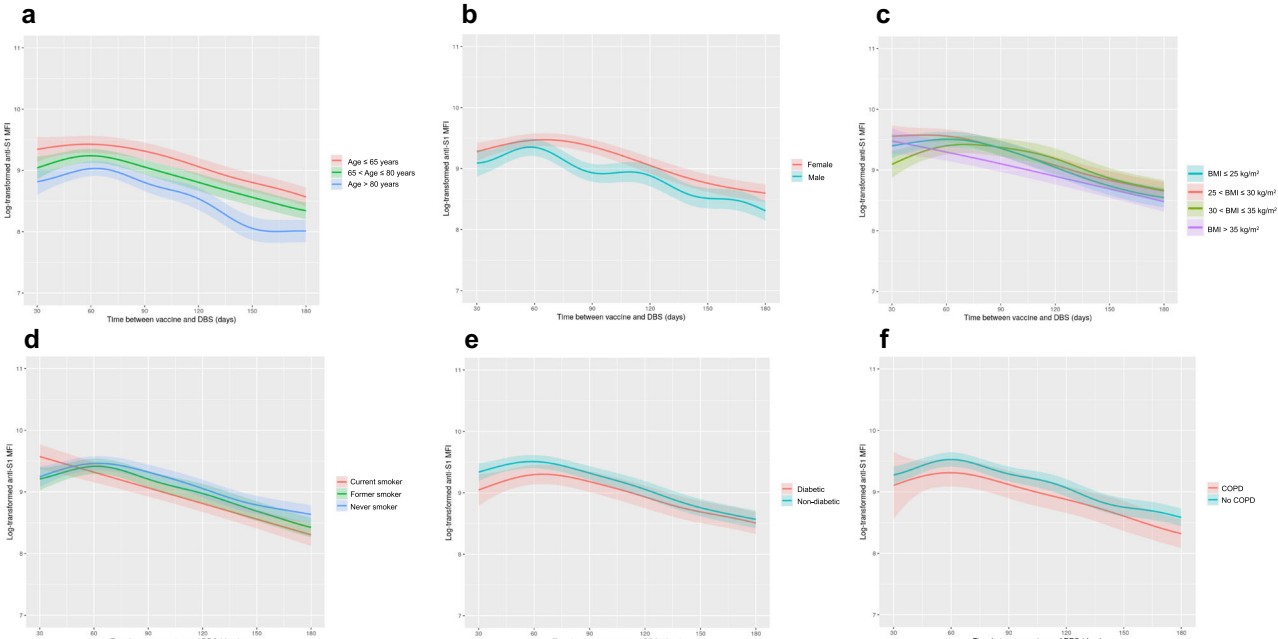

**Fig. 2 | Anti-S1 IgG levels over time between first vaccine dose and serosurvey by socio-demographic and clinical correlates.** Plot of predictited anti-S1 antibody levels by **A** age, **B** sex, **c** body mass index, **D** smoking history, **E** diabetes, and **F** chronic obstructive pulmonary disease. Predicted anti-S1 antibody levels generated from generalized additive model with smoothing spline since vaccination adjusted for age, sex, race/ethnicity, smoking history, education attainment, body mass index, diabetes, cardiovascular disease, asthma, chronic obstructive pulmonary disease, anti-N, COVID-19 infection history, dried blood spot batch, and vaccine type, except for those risk factors of primary interest that are stratified (**A**–**F**). Lighter color bands indicate 95% confidence intervals.

observed in our sample may also be attributable to the fact that C4R participants are older and have more comorbidities linked to immunosenescence. Another important distinction between data collected in C4R vs. U.K. cohorts relates to the timing of serological assessments: published U.K. studies occurred primarily in 2021, while C4R results span from mid-2021 to mid-2022.

Our work confirms that antibody responses are higher in persons with a history of both vaccination and natural infection, or "hybrid" immunity. History of hospitalized infection was associated with the highest antibody levels, followed by non-hospitalized infection prior to vaccination. Non-hospitalized infection following vaccination ("breakthrough") was associated with higher antibody responses versus non-infection in unadjusted analyses. The lack of statistical significance in the multivariable-adjusted analyses may be due to the relatively small number of post-vaccination infections in the sample, and thus limited power. The stronger association of hospitalized versus non-hospitalized infection with antibody response is consistent with prior literature showing augmented

antibody production following severe COVID illness[18–20]. Of importance, prior literature has also found that a subset of severely affected patients mount lower antibody responses, which has been hypothesized to contribute to impaired clearance of and recovery from SARS-CoV-2 in some individuals. Associations between post-vaccination and post-infection antibody responses and post-acute sequelae of SARS-CoV-2 infection (PASC) warrant further investigation.

Many of the factors associated with lower anti-S1 IgG MFI in this report are known to increase risk of severe acute COVID-19[21–23], and are also known to impact innate immune system dysfunction which is implicated as a risk factor for SARS-CoV-2 infection and poor clinical trajectories[24]. Our findings support the hypothesis that the associations of older age, male sex, obesity, diabetes and COPD with more severe acute COVID-19 could be partially mediated by an impaired immune response to SARS-CoV-2. Clinical trials and observational studies, including ours, have confirmed that additional vaccine doses are associated with higher antibody responses[25,26]. Higher anti-S1

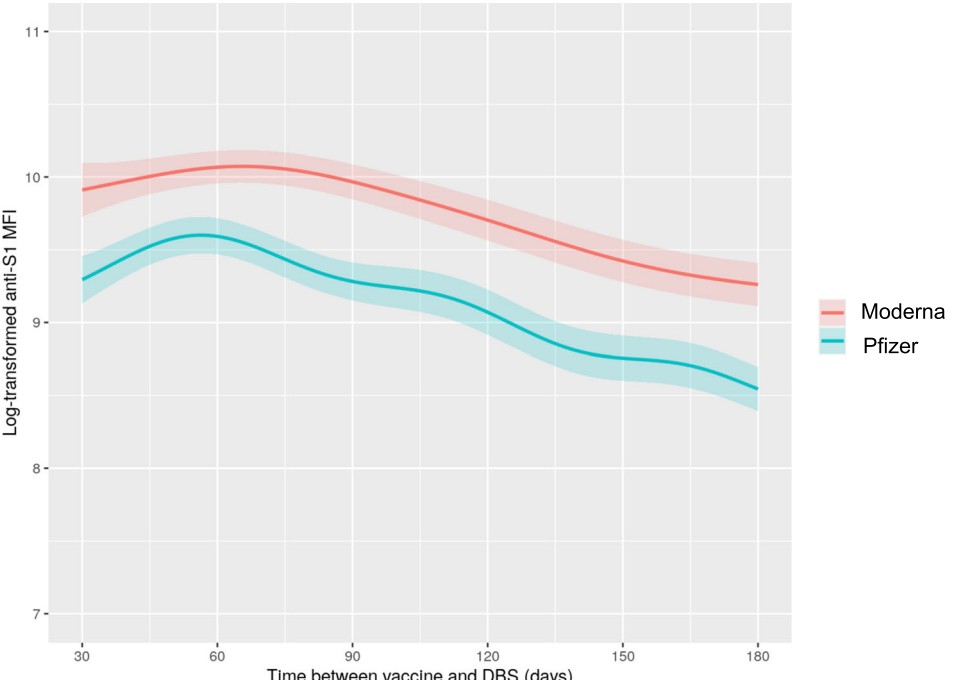

**Fig. 3 | Anti-S1 IgG levels over time between first vaccine dose and dried blood spot collection by receipt of BNT162b2 mRNA (Pfizer-BioNTech) or mRNA-1273 (Moderna) vaccine.** Predicted anti-S1 antibody levels generated from generalized additive model with smoothing spline since vaccination adjusted for age, sex, race/ ethnicity, smoking history, education attainment, body mass index, diabetes, cardiovascular disease, asthma, chronic obstructive pulmonary disease, anti-N, COVID-19 infection history, and dried blood spot batch. Lighter color bands indicate 95% confidence intervals.

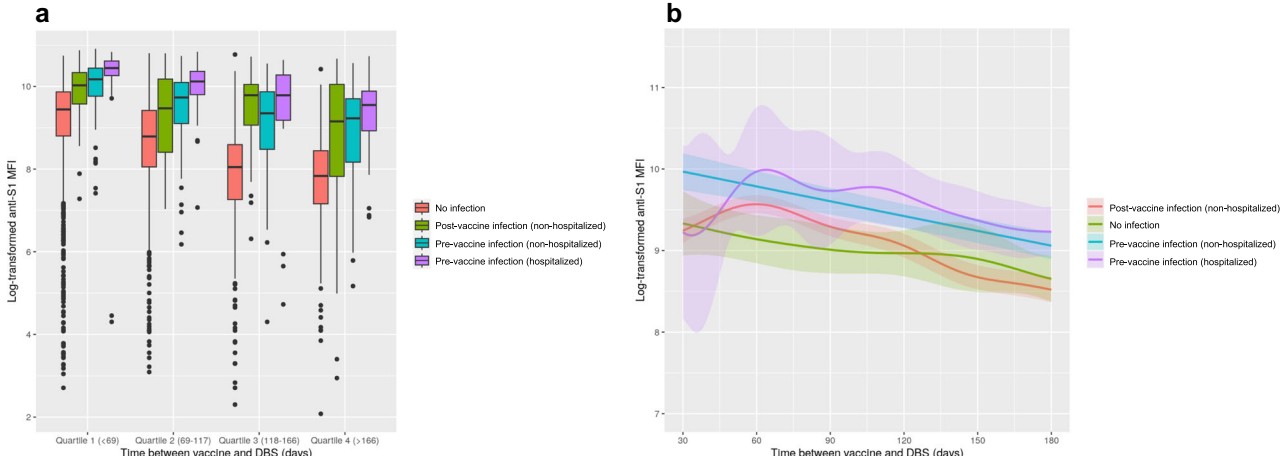

**Fig. 4 | Anti-S1 IgG levels over time between first vaccine dose and serosurvey by severity of acute infection. A** Unadjusted anti-S1 IgG levels by acute infection severity and time between first vaccine dose and dried blood spot collection categorized by quartiles in complete case cohort of participants with infection history and anti-S1 levels ($n = 3995$). Quartile 1 (no infection [$n = 1298$], non-hospitalized post-vaccine infection [$n = 59$], non-hospitalized pre-vaccine infection [$n = 68$], hospitalized pre-vaccine infection [$n = 18$]), Quartile 2 (no infection [$n = 1,069$], non-hospitalized post-vaccine infection [$n = 47$], non-hospitalized pre-vaccine infection [$n = 97$], hospitalized pre-vaccine infection [$n = 20$]), Quartile 3 (no infection [$n = 620$], non-hospitalized post-vaccine infection [$n = 52$], non-hospitalized pre-vaccine infection [$n = 119$], hospitalized pre-vaccine infection [$n = 32$]), Quartile 4 (no infection [$n = 294$], non-hospitalized post-vaccine infection [$n = 57$], non-hospitalized pre-vaccine infection [$n = 119$], hospitalized pre-vaccine infection [$n = 26$]). **B** Predicted anti-S1 antibody levels by severity of acute infection generated from generalized additive model with smoothing spline since vaccination adjusted for age, sex, race/ethnicity, smoking history, education attainment, body mass index, diabetes, cardiovascular disease, asthma, chronic obstructive pulmonary disease, anti-N, dried blood spot batch, and vaccine type ($n = 6,245$ from multiple imputed cohort). No infection ($n = 4823$), non-hospitalized post-vaccine infection ($n = 298$), non-hospitalized pre-vaccine infection ($n = 910$), hospitalized pre-vaccine infection $n = 214$). For Fig. 4A, the bounds of the box shows the 25th and 75th percentiles, thick black line indicates the median, the vertical upper and lower whiskers extended to the largest observed data point that falls within the 1.5 × interquartile range. Black dots indicate outlier values. For Fig. 4B, lighter color bands indicate 95% confidence intervals.

responses to vaccines have been shown to be correlated with a lower risk of future infection[9]. Taken together, our results suggest that additional (or higher) vaccine doses should be particularly encouraged for individuals at risk of lower anti-S1 responses to vaccination, including older individuals and those with obesity, diabetes, and/or COPD[27–29]. Future research will be important to determine if more frequent vaccination or higher dosing in these vulnerable groups are effective in preventing severe COVID-19.

## Strengths and limitations

Strengths of our study include a prospective, highly characterized, multi-ethnic, US general population-based sample that has comprehensive pre-pandemic phenotyping; semi-quantitative assessment of anti-S1 IgG responses using validated methods; an integrated definition of prior infection using self-report, hospital records, and anti-N antibody response; and extended follow-up after vaccination. Nonetheless, there are several limitations. First, information on neutralizing antibodies was not available, nor were other immunoglobulin isotypes (IgM, IgA) or measures of cellular immunity examined. Second, almost all of our participants had antibody levels above the threshold for "reactivity"[30] and it is unknown to what extent variation of antibody levels above that threshold is clinically meaningful. In future work, we intend to test if differences in anti-S1 IgG levels are associated with prospective risk of infection severity and PASC. Third, we did not have repeated antibody measures to examine within-person temporal trends in anti-S1 levels; nonetheless, our findings regarding antibody trends over time since vaccination resemble those from longitudinal vaccine trials with repeated measures. For example, although we only captured the date of the first vaccine dose, the observed antibody peak was 30-40 days following the anticipated date of the second vaccine dose, which is consistent with prior literature[5–7]. Fourth, for interpretability, our analysis was limited to participants who had received two doses of an approved vaccine. The dynamics and clinical significance of antibody levels in unvaccinated individuals and in individuals with three or more doses remain important areas for further investigation. Fifth, certain time-varying risk factors for differential antibody response may be misclassified. Clinical conditions, and smoking status, were measured several years prior to the serosurvey, which may conservatively bias our estimates. Infection history was defined primarily by self-report and could be subject to false-positive or false-negative classification, with uncertain influence on our estimates. We did not have data regarding health-related behaviors (e.g., masking and participation in large gatherings) that could correlate with probability of subclinical, unreported infection. Nonetheless, our models included anti-nucleocapsid antibody levels, which are an objective, sensitive biomarker of potential prior clinical or subclinical infection among vaccinated populations[31]. Results in a sensitivity analysis limited to participants with a history of a positive COVID-19 test yielded similar results. Of note, the measures used in this study were completed prior to the emergence of the Omicron variants, which may demonstrate different immunogenicity compared to earlier variants; this is another important area for investigation in future work. Sixth, the meta-cohort was not directly representative of the U.S. population and is subject to healthy volunteer bias, which likely biased our results towards the null and limits generalizability to groups that were relatively underrepresented, including younger adults and minoritized groups. Overall, the cohort was older and a significant proportion had co-morbidities. While adverse associations with advanced age were clearly shown, no consistent associations with race or ethnicity were observed, with the exception of higher and lower antibody levels in participants reporting Asian and AIAN, respectively, versus non-Hispanic White race. These subgroups were small, hence the finding should be interpreted with caution, but it is somewhat consistent with a prior study on higher antibody levels in Asian participants[32]. Finally, cohort heterogeneity could influence our results, yet C4R ascertained COVID-related exposures using standardized measures[16], pre-pandemic characteristics were harmonized following established protocols, and cohort-adjusted and cohort-stratified sensitivity analyses yielded highly consistent results.

## Conclusions

We identified differential anti-S1 antibody responses related to vaccine-related, socio-demographic, and clinical factors in a diverse U.S. population sample. Our findings might help to identify subgroups of adults who benefit from more frequent vaccination or other COVID-19 prevention strategies. Strategies to optimize COVID-19 vaccine responsiveness at the individual level, and deployment at the population level, warrant further research and development.

## Methods

### Study Participants

To study the impact of the COVID-19 pandemic on US adults, C4R aimed to perform standardized assessment of COVID-19 in participants in 14 longstanding NIH-funded prospective cohort studies (see Supplementary Methods)[16]. All cohort participants who were alive on March 1, 2020, and had not withdrawn consent for cohort participation, were considered eligible for C4R enrollment. Institutional review board approval was obtained from all study sites. Informed consent was obtained from each study participant.

Inclusion criteria for the present analysis were self-report of two mRNA vaccinations for COVID-19 (BNT162b2 mRNA or mRNA-1273) prior to submission of a valid dried blood spot (DBS) for serology assay. We excluded participants who reported only one vaccination prior to serosurvey completion, or were eligible for a third vaccination prior to their date of serosurvey completion (Supplementary Fig. 1).

### C4R serosurvey

The C4R serosurvey was accomplished via DBS, as previously described[16,30]. Briefly, the DBS requires that several drops of whole blood, from a finger prick or blood collection tube, be absorbed into a specially designed card. Participants who consented to the serosurvey completed the DBS at home or at an in-person exam. DBS samples were shipped to the central laboratory at the University of Vermont. The present study includes DBS samples collected between February 2021 and August 2022.

Serology assays were performed on DBS eluates by the Wadsworth Center, New York State Department of Health (Albany, NY, USA), using validated methods[30]. The assays were designed to detect IgG for SARS-CoV-2 S1 protein, which may be induced by natural infection or currently approved COVID-19 vaccines, and nucleocapsid (N) protein, which is induced by natural infection only. Briefly, SARS-CoV-2 S1 and N antigens (Sino Biological, Wayne, PA, USA) were covalently coupled to Magplex-C microspheres (Luminex Corp., Austin, TX, USA) with different bead regions coupled to the SARS-CoV-2 S1 and N antigens (Sino Biological, Wayne, PA, USA). Median fluorescence intensity (MFI) was analyzed using a FlexMap 3D instrument (Luminex Corp., Austin, TX, USA). Five separate bead sets were used over the period of analysis.

Antibody response was classified as reactive or non-reactive based on the mean and standard deviation (SD) of anti-S1 MFI values in uninfected (pre-pandemic) DBS samples[30]. For each bead set, the reactivity threshold was calculated as the mean + 6 SD of the uninfected (pre-pandemic) MFI. Samples above this threshold were classified as reactive.

### C4R questionnaires

C4R collected information on SARS-CoV-2 infection and vaccination status via two waves of questionnaires conducted from April 2020 through February 2023; the current report includes data from questionnaires collected through August 2022. Questionnaires were administered by telephone interview, electronic survey, in-person examination, and/or mailed pamphlet. Participants were asked about history of SARS-CoV-2 infection, SARS-CoV-2 testing, COVID-19 hospitalization, COVID-19 vaccination status, date of first vaccine administration, number of vaccines received, and vaccine manufacturer. Infections were dichotomized according to whether they were, or were not, associated with hospitalization. Definite history of SARS-CoV-2 infection was classified based on self-report of a positive SARS-CoV-2 test, adjudication of medical records for a COVID-19 hospitalization, or evidence of anti-nucleocapsid antibody reactivity on DBS; self-

reported infections not meeting any of these criteria were classified as probable (Supplementary Table 7). Both definite and probable cases of infection were included in the main analyses, and a sensitivity analysis was performed that excluded probable cases.

## Pre-pandemic measures

C4R cohorts have performed longitudinal data collection on participants for up to 51 years. C4R harmonized these data across cohorts, as previously described[16]. Age, sex, and educational attainment were self-reported. Race and ethnicity, which were self-reported and categorized according to the 2000 Census methods[33,34], were included in this study to address specific knowledge gaps about disparities in COVID-19 outcomes among members of historically marginalized or underserved populations. Time-varying risk factors were defined using the most recently collected data by each cohort. Smoking status was self-reported as never, former or current. Height, weight, blood pressure, fasting lipids and blood glucose were measured using standardized protocols. Hypertension was defined as a systolic blood pressure≥140 mm Hg, diastolic blood pressure≥90 mm Hg, or antihypertensive medication use. Diabetes was defined as fasting blood glucose≥126 mg/dL or use of insulin or hypoglycemic medications. Chronic kidney disease was defined as estimated glomerular filtration rate (eGFR) <45 mL/min/1.73m$^2$. Cardiovascular disease, asthma, and chronic obstructive pulmonary disease (COPD) were identified via self-report or by the ascertainment of relevant clinical events, confirmed by medical record review, over cohort follow-up.

## Statistical analysis

Natural log-transformed anti-S1 IgG level was regressed on candidate risk factors, time-since-vaccination (defined as days from first vaccination to serosurvey), and laboratory batch, using linear regression. Associations with candidate risk factors were tested with and without multivariable adjustment for other candidate risk factors under consideration. For interpretability purposes, we exponentiated the beta coefficient and presented the results as mean percent difference in anti-S1 level in the comparison vs. reference group.

Generalized additive models were used to examine differences in log-transformed anti-S1 IgG level over time-since-vaccination. Based on prior literature and empiric data in this study[5,7], we fitted linear splines for the periods before and after the peak antibody level at 60 days-since-vaccination. Using the linear spline model, predicted antibody levels were plotted over time by risk factor strata. To test associations of candidate risk factors with rate of antibody waning during the second (post-60 day) period, we assessed the beta coefficient for the interaction term, "risk factor×slope of time from vaccine to serosurvey (per 30 days)." The beta coefficient was exponentiated with positive coefficients indicating slower decline, and negative coefficients more rapid decline, in anti-S1 IgG levels.

Multiple imputation by chained equations was applied to handle covariate missing data (Supplementary Table 8). Ten imputed datasets were created, and Rubin's rule was applied to combine the estimates from the imputed datasets[35]. Characteristics of complete cases were similar to the imputed (primary) dataset (Supplementary Table 9). Complete case analysis was performed as a sensitivity analysis.

To explore whether inclusion of a certain cohort or set of cohorts was influential with respect to the overall results, we repeated the main analysis using a "leave-one-out" approach by separately removing participants in the three largest cohorts in this analysis (ARIC, FHS, and REGARDS) from the pooled analysis sample. We also performed a sensitivity analysis in which we only included participants from ARIC, FHS, and REGARDS. Analyses were also repeated with exclusion of participants reporting only one dose of vaccine. Results from all subgroup analyses should be interpreted with caution due to the possibility for type 1 or type 2 error.

All analyses were conducted in R (R Statistical Foundation, Vienna, Austria)[16]. R packages used for analysis include: "tidyverse", "ggplot2", "arsenal", "mice", "mgcv", and "lspline". Two-sided $p$-values < 0.05 were considered statistically significant.

## Reporting summary

Further information on research design is available in the Nature Portfolio Reporting Summary linked to this article.

## Data availability

All data supporting the findings described in this manuscript are available in the article and in the Supplementary Information. Data harmonization for C4R performed using SAS Studio (SAS Data Science) in Seven Bridges Platform. Raw deidentified data is available, upon request and with appropriate consortium and cohort permissions, on the C4R Analysis Commons. Preliminary responses to data requests will be made within 4 weeks of receipt. C4R Analysis Commons, hosted on BioData Catalyst powered by Seven Bridges (https://accounts.sb.biodatacatalyst.nhlbi.nih.gov/). Further information also available at the C4R website: https://c4r-nih.org.

## Code availability

Code for data cleaning and analysis is available, upon request and with appropriate consortium and cohort permissions, on the C4R Analysis Commons.

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

## Acknowledgements

The Collaborative Cohort of Cohorts for COVID-19 Research (C4R) Study is supported by National Heart, Lung, and Blood Institute (NHLBI)–Collaborating Network of Networks for Evaluating COVID-19 and Therapeutic Strategies (CONNECTS) grant OT2HL156812, with cofunding from the National Institute of Neurological Disorders and Stroke (NINDS) and the National Institute on Aging (NIA). The Atherosclerosis Risk in Communities Study has been funded in whole or in part by the NHLBI, National Institutes of Health (NIH), US Department of Health and Human Services, under contracts 75N92022D00001, 75N92022D00002, 75N92022D00003, 75N92022D00004, and 75N92022D00005. Neurocognitive data are collected under grants U01 2U01HL096812, 2U01HL096814, 2U01HL096899, 2U01HL096902, and 2U01HL096917 from the NHLBI, the NINDS, the NIA, and the National Institute on Deafness and Other Communication Disorders. Ancillary studies funded additional data elements. The Blood Pressure and Cognition Study is supported by the NINDS (grant R01 NS102715). The Coronary Artery Risk Development in Young Adults (CARDIA) Study is conducted and supported by the NHLBI in collaboration with the University of Alabama at Birmingham (contracts HHSN268201800005I and HHSN268201800007I), Northwestern University (contract HHSN268201800003I), the University of Minnesota (contract HHSN268201800006I), and the Kaiser Foundation Research Institute (contract HSN268201800004I). The Genetic Epidemiology of COPD (COPDGene) Study was supported by awards U01 HL089897 and U01 HL089856 from the NHLBI. COPDGene is also supported by the COPD Foundation through contributions made to an industry advisory board comprised of AstraZeneca AB (Cambridge, United Kingdom), Boehringer-Ingelheim (Ingelheim am Rhein, Germany), Genentech, Inc. (South San Francisco, California), GlaxoSmithKline plc (London, United Kingdom), Novartis International AG (Basel, Switzerland), Pfizer, Inc. (New York, New York), Siemens AG (Berlin, Germany), and Sunovion Pharmaceuticals Inc. (Marlborough, Massachusetts). The Framingham Heart Study has received support from the NHLBI (grant N01-HC-25195, contract HHSN268201500001I, and grant 75N92019D00031). The Hispanic Community Health Study/Study of Latinos (HCHS/SOL) is a collaborative study supported by contracts between the NHLBI and the University of North Carolina (contract HHSN268201300001I/N01-HC-65233), the University of Miami (contract HHSN268201300004I/N01-HC-65234), Albert Einstein College of Medicine (contract HHSN268201300002I/N01-HC-65235), the University of Illinois at Chicago (contract HHSN268201300003I/N01-HC-65236 (Northwestern University)), and San Diego State University (contract HHSN268201300005I/N01-HC-65237). The following institutes/centers/offices have contributed to the HCHS/SOL through a transfer of funds to the NHLBI: the National Institute on Minority Health and Health Disparities, the National Institute on Deafness and Other Communication Disorders, the National Institute of Dental and Craniofacial Research, the National Institute of Diabetes and Digestive and Kidney Diseases, the NINDS, and the NIH Office of Dietary Supplements. The Jackson Heart Study is supported by and conducted in collaboration with Jackson State University (contract HHSN268201800013I), Tougaloo College (contract HHSN268201800014I), the Mississippi State Department of Health (contract HHSN268201800015I), the University of Mississippi Medical Center (contracts HHSN268201800010I, HHSN268201800011I, and HHSN268201800012I), the NHLBI, and the National Institute on Minority Health and Health Disparities. The Mediators of Atherosclerosis in South Asians Living in America (MASALA) Study was supported by grant R01HL093009 from the NHLBI, the National Center for Research Resources, and the National Center for Advancing Translational Sciences, NIH, through University of California, San Francisco–Clinical and Translational Science Institute grant UL1RR024131. The Multi-Ethnic Study of Atherosclerosis (MESA) and the MESA SNP Health Association Resource (SHARe) are conducted and supported by the NHLBI in collaboration with the MESA investigators. Support for MESA is provided by

grants and contracts 75N92020D00001, HHSN268201500003I, N01-HC-95159, 75N92020D00005, N01-HC-95160, 75N92020D00002, N01-HC-95161, 75N92020D00003, N01-HC-95162, 75N92020D00006, N01-HC-95163, 75N92020D00004, N01-HC-95164, 75N92020D00007, N01-HC-95165, N01-HC-95166, N01-HC-95167, N01-HC-95168, N01-HC-95169, R01-HL077612, R01-HL093081, R01-HL130506, R01-HL127028, R01-HL127659, R01-HL098433, R01-HL101250, and R01-HL135009 from the NHLBI; grant R01-AG058969 from the NIA; and grants UL1-TR-000040, UL1-TR-001079, and UL1-TR-001420 from the National Center for Advancing Translational Sciences. Funding for SHARe genotyping was provided by NHLBI contract N02-HL-64278. This publication was developed under Science to Achieve Results (STAR) research assistance agreements RD831697 (MESA Air) and RD-83830001 (MESA Air Next Stage), awarded by the Environmental Protection Agency. Whole genome sequencing for the Trans-Omics in Precision Medicine (TOPMed) Program was supported by the NHLBI. Whole genome sequencing for the MESA component of the TOPMed Study (Database of Genotypes and Phenotypes accession no. phs001416.v1.p1) was performed at the Broad Institute of MIT and Harvard (grant 3U54HG003067-13S1). Centralized read mapping and genotype calling, along with variant quality metrics and filtering, were provided by the TOPMed Informatics Research Center (grant 3R01HL-117626-02S1 and contract HHSN268201800002I) (Broad RNA Seq, Proteomics HHSN268201600034I, UW RNA Seq HHSN268201600032I, USC DNA Methylation. HHSN268201600034I, Broad Metabolomics HHSN268201600038I). Phenotype harmonization, data management, sample-identity quality control, and general study coordination were provided by the TOPMed Data Coordinating Center (grants 3R01HL-120393 and U01HL-120393 and contract HHSN268180001I). The provision of genotyping data was supported in part by the National Center for Advancing Translational Sciences, Clinical and Translational Science Institute grant UL1TR001881, and National Institute of Diabetes and Digestive and Kidney Diseases Diabetes Research Center grant DK063491 to the Southern California Diabetes Endocrinology Research Center. The NHLBI Pooled Cohorts Study was supported by grants R21HL153700, K23HL130627, R21HL129924, and R21HL121457 from the NIH/NHLBI. The Northern Manhattan Study was supported by grants R01 NS29993 and R01 NS48134 from the NINDS and grant R01 AG066162 from the NIA. The Prevent Pulmonary Fibrosis cohort study was established in 2000 and has been supported by NIH awards Z01-ES101947, R01-HL095393, RC2-HL1011715, R21/33-HL120770, R01-HL097163, Z01-HL134585, UH2/3-HL123442, P01-HL092870, UG3/UH3-HL151865, and DoD W81XWH-17-1-0597. The Reasons for Geographic and Racial Differences in Stroke (REGARDS) Study is supported by cooperative agreement U01 NS041588, cofunded by the NINDS and the NIA. Research by the principal and co–principal investigators of the Severe Asthma Research Program was funded by the NIH/NHLBI (grants U10 HL109164, U10 HL109257, U10 HL109146, U10 HL109172, U10 HL109250, U10 HL109168, U10 HL109152, and U10 HL109086). Additional support was provided through industry partnerships with the following companies: AstraZeneca, Boehringer-Ingelheim, Genentech, GlaxoSmithKline, MedImmune, Inc. (Gaithersburg, Maryland), Novartis, Regeneron Pharmaceuticals, Inc. (Tarrytown, New York), Sanofi S.A. (Paris, France), and Teva Pharmaceuticals USA (North Wales, Pennsylvania). Spirometers used in Severe Asthma Research Program III were provided by nSpire Health, Inc. (Longmont, Colorado). The Subpopulations and Intermediate Outcome Measures in COPD Study (SPIROMICS) has been funded by contracts with the NIH/NHLBI (contracts HHSN268200900013C, HHSN268200900014C, HHSN268200900015C, HHSN268200900016C, HHSN268200900017C, HHSN268200900018C, HHSN268200900019C, and HHSN268200900020C) and grants from the NIH/NHLBI (grants U01 HL137880 and U24 HL141762) and supplemented through contributions made to the Foundation for the NIH and the COPD Foundation by AstraZeneca, MedImmune, Bayer Corporation (Whippany, New Jersey), Bellerophon Therapeutics (Warren, New Jersey), Boehringer-Ingelheim, Chiesi Farmaceutici S.p.A. (Parma, Italia), the Forest Research Institute, Inc. (Jersey City, New Jersey), GlaxoSmithKline, Grifols Therapeutics, Inc. (Research Triangle Park, North Carolina), Ikaria, Inc. (Hampton, New Jersey), Novartis, Nycomed Pharma GmbH (Zurich, Switzerland), ProterixBio, Inc. (Billerica, Massachusetts), Regeneron, Sanofi, Sunovion, Takeda Pharmaceutical Company (Tokyo, Japan), Theravance Biopharma, Inc. (South San Francisco, California), and Mylan N.V. (White Sulphur Springs, West Virginia). The Strong Heart Study has been funded in whole or in part by the NHLBI (contracts 75N92019D00027, 75N92019D00028, 75N92019D00029, and 75N92019D00030). The Strong Heart Study was previously supported by research grants R01HL109315, R01HL109301, R01HL109284, R01HL109282, and R01HL109319 and cooperative agreements U01HL41642, U01HL41652, U01HL41654, U01HL65520, and U01HL65521. NIH U01CA260508 (N.J.M.), NIH R01HL168126 (J.S.L.), NIH K23HL150301 (J.S.K.).

The views expressed in this manuscript are those of the authors and do not necessarily represent the views of the National Heart, Lung, and Blood Institute; the National Institutes of Health; or the U.S. Department of Health and Human Services.

## Author contributions

J.S.K, Y.S., P.B., R.T.D., E.C.O., acquired and interpreted the data, wrote the initial draft of the manuscript, critical revision of the manuscript for important intellectual content. L.M.S., N.J.M., and M.M.P. generated serosurvey data. R.P.T., R.T.D, and E.C.O. obtained funding. M.C., R.B., R.P.T., L.M.S., T.D.B., M.R.A., N.B.A., P.J.S., R.P.B., D.A.S., J.S.L., V.X., M.D., E.A.R., B.J.M., A.M.K., S.E.W., J.C., C.R.I., L.M.R., M.S.V.E., V.J.H., V.E.O., P.W., S.A.C., J.M.H., N.J.M., M.M.P., reviewed and interpreted the data and contributed to the writing of the manuscript. R.T.D. and E.C.O. provided supervision.

## Competing interests

J.S.L. reports grants from the NIH and Boehringer Ingelheim, outside the submitted work, has received an unrestricted research gift from Pliant, outside the submitted work, consulting fees from Blade, Boehringer Ingelheim, United Therapeutics, Astra Zenca, and Eleven P15, outside the submitted work, has served on data safety monitoring board for United Therapeutics and Avalyn, and is an advisor for the Pulmonary Fibrosis Foundation, outside the submitted work. The remaining authors declare no competing intrests.

## Additional information

[1]Department of Medicine, University of Virginia School of Medicine, Charlottesville, VA, USA. [2]Department of Medicine, Columbia University Vagelos College of Physicians and Surgeons, New York, NY, USA. [3]Department of Biostatistics, Columbia University Mailman School of Public Health, New York, NY, USA. [4]Department of Medicine, Larner College of Medicine at the University of Vermont, Burlington, VT, USA. [5]Department of Pathology and Laboratory Medicine, Larner College of Medicine at the University of Vermont, Burlington, VT, USA. [6]Division of Infectious Diseases, Wadsworth Center, New York State Department of Health, Albany, NY, USA. [7]Department of Medicine, University of Pennsylvania, Philadelphia, PA, USA. [8]Department of Preventive Medicine, Northwestern University Feinberg School of Medicine, Chicago, IL, USA. [9]Division of Epidemiology and Community Health, School of Public Health, University of Minnesota, Minneapolis, MN, USA. [10]Division of Pulmonary, Critical Care and Sleep Medicine, National Jewish Health, Denver, CO, USA. [11]Department of Medicine, University of Colorado School of Medicine, Aurora, CO, USA. [12]Department of Medicine, Boston University Chobanian and Avedisian School of Medicine, Boston, MA, USA. [13]Framingham Heart Study, Framingham, MA, USA. [14]Division of Rheumatology, National Jewish Health, Denver, CO, USA. [15]Division of General Internal Medicine, University of California San Francisco, San Francisco, CA, USA. [16]Department of Medicine, Department of Immunology, and Department of Environmental Medicine and Occupational Health, University of Pittsburgh School of Medicine, School of Public Health, Pittsburgh, PA, USA. [17]Department of Population Health, New York University Grossman School of Medicine, New York University Langone Health, New York, NY, USA. [18]Department of Medicine, New York University Grossman School of Medicine, New York University Langone Health, New York, NY, USA. [19]Department of Epidemiology and Population Health, Albert Einstein College of Medicine, Bronx, NY, USA. [20]Department of Genetics, University of North Carolina, Chapel Hill, NC, USA. [21]Department of Neurology, Columbia University Vagelos College of Physicians and Surgeons, New York, NY, USA. [22]Department of Epidemiology, Columbia University Mailman School of Public Health, New York, NY, USA. [23]Department of Epidemiology, School of Public Health, University of Alabama at Birmingham, Birmingham, AL, USA. [24]Division of Respiratory Medicine, Mayo Clinic, Scottsdale, AZ, USA. [25]Division of Pulmonary and Critical Care Medicine, University of California San Francisco, San Francisco, CA, USA. [26]Population Health Program, Texas Biomedical Research Institute, San Antonio, TX, USA. [27]Department of Pathology and Laboratory Medicine, Boston University Chobanian & Avedisian School of Medicine and Boston Medical Center, Boston, MA, USA. [28]Department of Biomedical Sciences, School of Public Health, University at Albany, Albany, NY, USA. [29]Division of Epidemiology, Department of Quantitative Health Sciences, College of Medicine and Science, Mayo Clinic, Rochester, MN, USA. [30]These authors contributed equally: Ryan T. Demmer, Elizabeth C. Oelsner. ✉e-mail: demmer.ryan@mayo.edu; eco7@cumc.columbia.edu

