## [Peer Review File · Nature Communications]

Demographic and Clinical Factors Associated With SARS-CoV-2 Spike 1 Antibody Response Among Vaccinated US Adults: the C4R StudyReviewers' Comments:

Reviewer #1:

Remarks to the Author:

As a medical epidemiologist mainly involved in vaccine effectiveness studies, I have commented on the objective, chosen outcome, study design, analysis and related discussion. My understanding is that further detailed laboratory and immunological aspects will be also reviewed by related specialists. This study is a highly relevant contribution to the identification of risk factors for lower anti-S1 antibody response and decrease over time after COVID-19 vaccination. Such studies, in complement to post marketing vaccine effectiveness studies, help to understand duration of protection and to identify subpopulation who could potentially benefit from adapted vaccination schedules. The study is based on the pooled analysis of 14 cohorts that is described as a collaborative cohort of cohorts representative of the US population.

Comments

The objective of the study is to identify risk factors for lower level of anti-S1 IgG antibody response and decline over time since vaccination in a US population based sample. However, authors also use higher levels as the described outcome. This sometimes affects the otherwise very logical flow of the manuscript.

Authors are using a non-random convenient sample from a pool of 14 cohorts included the Collaborative Cohort of Cohorts for COVID Research (C4R). In figure 1, authors clearly describe the exclusion criteria from the 49 500 persons included in C4R data set to the 7 797 included in the current analysis. Since the sample is not a random sample, it would be useful if authors could discuss the potential consequences of exclusions (e.g. 31197 who did not provide DBS sample). For representativeness purpose, were authors able to compare characteristics (potential risk factors) of the 6797 participants to the rest of the 49 500 cohorts members? This in order to identify potential selection biases and their consequences on outcome measurement. In addition, authors may want to discuss (compared to US population) the effect of a follow up cohort (in which each participant is under observation) on the health behaviour changes over time if any.

Authors have cited heterogeneity as an issue potentially taken care of by standardised protocols between cohorts. Could authors provide results of tests for heterogeneity and the potential consequences on data analysis? In addition, my understanding of table 1 is that three of the cohorts (ARIC, FHS and REGARDS) contribute to around 80% of the pooled sample. Are results different between each of the three large cohorts, and between those three large cohorts and the 11 other smaller cohorts? Is there an added value for the inclusion of the 11 smaller cohorts?

Authors suggest that variation in vaccine-induced antibody response may be clinically significant since lower humoral responses to SARS-CoV-2 vaccines are predictive of higher risk for breakthrough infections and severe COVID-19 -2 clinical outcome. Authors suggest that using the mean percent difference in anti-S1 MFI change provided a simple and clearer understanding of change in antibody levels. They may want to discuss, (beside statistical significance) the clinical and public health meaning of a specific, sometimes low, percent change in anti-S1 MFI.

Due to missing values authors did an imputation. Could they provide details of the imputation method? In addition, in some occurrences (prior infection) the numbers of participants included in the respective analyses (imputed and not) are quite different and the validity of an imputation may be discussed.

In eTable for non imputed data, authors may want to show number and percent of missing information for each variable.

From Table 1, 9,1% of the participants only received one dose of vaccine. Do results differ when excluding them?

From Table 2 authors suggest that, as time between first vaccine dose and DBS lengthened, non-reactivity is more prevalent among BNT162b2 than mRNA-1273 recipients. However, table 2 (which shows reactivity), suggests very small absolute differences in non-reactivity prevalence.

One of the major contributing factors to the level of anti-S1 IgG antibody levels is the occurrence of prior SARS-CoV-2 infection. Authors mention that documentation of prior infection was based on self-

reporting and/or medical records review. Could authors describe what were the precise definitions of self reported (clinical? Self-tests? Etc.) and medically recorded (RT-PCR confirmed?) prior infection? Computation of sensitivity and specificity of definitions (and related predictive values) may be relevant when adjusting for or stratifying analysis by prior infection. It may also affect results according to time since prior infection.

In Figure 1 authors show anti-S1 MFI log transformed. As such, should not we expect a log Y axis instead of an arithmetic one? Same question for Figure 2.

Authors suggest that mRNA-1273 (Moderna) vaccine induces a higher antibody response than the BNT162b2mRNA (Pfizer-BioNTech). In their discussion authors may want to acknowledge the different dosages between the two vaccines and discuss the consequences.

Authors mention that anti-nucleocapsid antibody levels was associated with higher anti-S1 IgG. This is a difficult association to measure since anti-S1IgG measures both natural and vaccine induced antibodies.

Overall, this study can be a major contribution to understanding duration of protection after vaccination while taking prior infection into account. My main suggestion to authors is to further discuss the representativeness of the non-random study sample and the potential consequences on the results. The understanding of the current results would ideally be complemented by an analysis of antibody levels and decline after infection in the unvaccinated C4R population.

Reviewer #2:

Remarks to the Author:

This study is an analysis of risk factors for lower anti-S1 antibody response following COVID-19 vaccination from multiple population-based cohorts of adults in the USA. The study authors suggest that the magnitude and durability of antibody levels, has not been comprehensively investigated in multi-ethnic, diverse, population-based U.S. samples previously, but acknowledge there are other studies globally that have included large numbers of mRNA vaccine receipts including several from the UK.

I have several suggestions which I hope are constructive and aimed at improving the paper.

The current title uses the word "predictors" but I think this should be removed and replaced with, for example, association, as this is not a predictive study design, it is observational in nature.

It would be helpful if more background information on the cohorts and their representativeness as a general population sample was provided to aid readers of this paper as a standalone article, rather than having to refer to external literature.

My main concern with the analysis at present is regarding the lack of stratification of participants post first and second dose vaccine in analyses, particularly tables 1-3 and figures 1-5. There is good evidence of a quantitative difference in S level response between first and second dose so to combine these into one group for these descriptions of the anti-S1 levels seems inappropriate. This approach is inconsistent with most previous analyses including for e.g. Wei, J., Stoesser, N., Matthews, P.C. et al. Antibody responses to SARS-CoV-2 vaccines in 45,965 adults from the general population of the United Kingdom. *Nat Microbiol* 6, 1140–1149 (2021). <https://doi.org/10.1038/s41564-021-00947-3>

Some additional information about the cohorts and how data were collected would be useful in terms of understanding risk of bias, including:

- How were cohorts originally sampled?
- How were diagnoses of COVID made?
- How was self-report data collected during the pandemic?

- How did the analyses account for across cohort heterogeneity?
- What is geographical spread of the cohorts?
- How were the cohorts followed up during covid and how did this compare to previous follow up methods?
- How was severe covid defined/classified?
- How much missing data was there across the cohorts and how was this dealt with?

Why was the group Novavax/AZ combined in table 1 and these individuals excluded elsewhere in the analysis?

Are the results in Table 3 univariate or multivariate with everything else in the table included in the model? If it's the later then this seems inappropriate.

Limitations should include a discussion on the higher proportion of females and older age profile of the combined cohort and issues regarding generalisability as a result of these biases.

Authors state that "UK Real-Time Assessment of Community Transmission-2 (REACT-2) study quantified antibody reactivity as a binary variable up to 20 weeks post-vaccination among a sample mostly receiving either BNT162b2 or ChAdOx1 vaccines." However they do not acknowledge that several other UK studies did use continuous measures including the UK ONS and Virus Watch studies.

"Consistent with prior literature, we observed that anti-S1 levels peaked at around 60 days after vaccination with a subsequent decline." This does not seem to be consistent with existing literature from the UK where both ONS and Virus Watch show a much earlier peak post first and second dose. The authors should re-examine this finding after splitting participant data into first and second dose responses.

"Strategies to optimize COVID-19 vaccine responsiveness at the individual level, and deployment at the population level, warrant further research and development. " Can the authors be more specific and draw specifically from their results in terms of which issues require further research and development.

Reviewer #3:

Comments to Manuscript: Predictors of SARS-CoV-2 Spike 1 Antibody Response Among Vaccinated US Adults: the C4R Study

This study investigated the risk factors associated with lower anti-S1 antibody response following COVID-19 vaccination in a U.S. population-based sample of adults. BNT162b2 vaccine, age > 80 years, female sex, and a number of other risk factors were found associated with higher anti-S1 antibody levels. BNT162b2 vaccine and age > 80 years were found associated with higher rate of antibody level decline.

- The study presents useful knowledge on antibody response after vaccination based on a “multi-ethnic, diverse, population-based U.S. samples”. However, the samples are mostly older U.S. adults participating in longitudinal cohort studies, that may not be representative of the general US population (e.g., line 342 “generalizability to the U.S. population”).
- Line 265: I’m not sure what do you mean by “As the time between first vaccine dose and DBS lengthened”. Non-reactivity was more prevalent among those who received BNT162b2 vaccine and those 80 years or older across all groups defined by similar time between first vaccine dose and DBS.
- Line 276: “ Asian race was associated with greater antibody response”. It should be compared to non-Hispanic white.
- Line 279: What is “ fully adjusted models”? Is the model used for health behaviors and comorbidities different from the rest of the variables. From your description of statistical analysis, it seems like there is only one model for antibody response (“Log-transformed anti-S1 MFI was regressed on candidate risk factors using multivariable linear regression models, adjusted for time-since-vaccination and laboratory batch.”)
- Line 279: “Lower antibody responses were observed in former smokers (6.7%, 95% CI, 0.9-12.2) and current smokers (15.2%, 95% CI, 5.9-23.7). ” Again, please provide the

reference group in the text.

- Line 284: “SARS-CoV-2 infection was strongly associated with higher anti-S1 IgG antibody response.” Should it be prior COVID-19 disease instead of SARS-CoV-2 infection? If “hospitalized” means “severe” and “non-hospitalized” means “not severe” in Table 3, then please be consistent in the text and the table.
- Line 302: What do you mean by “Based on the largest effect estimates from multivariable regression analysis”? Did you include only variables with the largest effect estimates?
- What is the percentage of missing covariables in your model?

Response to Reviewers for NCOMMS-23-19754

Reviewer #1 (R1)

R1.1a The objective of the study is to identify risk factors for lower level of anti-S1 IgG antibody response and decline over time since vaccination in a US population-based sample. However, authors also use higher levels as the described outcome. This sometimes affects the otherwise very logical flow of the manuscript.

Response: We have modified the text to improve clarity and consistency with respect to the directions of associations reported (higher versus lower anti-S1)

- *Changes throughout.*

R1.1b Authors are using a non-random convenient sample from a pool of 14 cohorts included the Collaborative Cohort of Cohorts for COVID Research (C4R). In figure 1, authors clearly describe the exclusion criteria from the 49 500 persons included in C4R data set to the 6797 included in the current analysis. Since the sample is not a random sample, it would be useful if authors could discuss the potential consequences of exclusions (e.g. 31197 who did not provide DBS sample). For representativeness purpose, were authors able to compare characteristics (potential risk factors) of the 6797 participants to the rest of the 49 500 cohorts members? This in order to identify potential selection biases and their consequences on outcome measurement.

Response: We agree that non-random sampling is a limitation to our report and have noted this in the limitations section. We have added a supplementary table that shows the baseline characteristics of participants who had vaccine data but were excluded from the sample due to missing dried blood spot (DBS) data (n=19546, **Supplementary Table 1**). Compared to our analytical sample, the group without DBS data had the following characteristics: i) younger age; ii) higher proportions of African American/Black and Hispanic participants; iii) higher proportion of former smokers; and iv) lower proportion of those who received the mRNA-1273 (Moderna) vaccine. Fundamentally, the primary concern in this setting is selection bias that is differential by both levels of exposure(s) and the S1 outcome. For example, preferentially over-selecting individuals who have reduced S1 response among the older but not younger individuals would bias parameter estimates for age towards inverse associations. Similar patterns of selection related to other variables could induce spurious results moving towards or away from the null. However, we have no reason to believe that the selection factors would lead to this kind of unique differential selection. As such, nondifferential selection bias, if it exists, would likely bias our results towards the null. In terms of external validity/generalizability, although our sample is not random or nationally representative, our sample does include individuals across the adult age spectrum, from each gender, and across multiple race/ethnicities. We have made the following additions to the manuscript in response to the reviewer's comment.

- *Results:* We have made significant changes throughout the results based on reviewer comments. In specific response to this critique, we have added Supplementary Table 1 showing "Baseline characteristics of C4R participants not included in the analysis due to missing serosurvey". We also note the following in the text: "Of note, compared to the participants eligible for this report, C4R participants who self-reported vaccination but did

not complete DBS were younger, more racially and ethnically diverse, and less likely to have received the mRNA-1273 vaccine (**Supplementary Table 1**).”

- *Discussion*: The following limitation text was added to the discussion: “Sixth, the meta-cohort was not directly representative of the US population and is subject to healthy volunteer bias, which likely biased our results towards the null and limits generalizability to groups that were relatively underrepresented, including younger adults and minoritized groups.”

R1.1c In addition, authors may want to discuss (compared to US population) the effect of a follow up cohort (in which each participant is under observation) on the health behaviour changes over time if any.

Response: We agree that cohorts which have been followed for many years are likely comprised of individuals who differ from the general population and often these cohorts exhibit healthy volunteer bias. This might have resulted in a higher prevalence of healthy behaviors related to COVID-19 risk mitigation as well (e.g., vaccination, masking). Indeed, the decision to be vaccinated was an eligibility criterion for this study. If unmeasured factors relating to healthy volunteer bias were associated with greater antibody response, this could have conservatively biased the associations we observed between older age and clinical comorbidities with lower antibody response. We have now acknowledged the limitation of health volunteer bias in the discussion.

- “Sixth, the meta-cohort was not directly representative of the US population and is subject to healthy volunteer bias, which likely biased our results towards the null and limits generalizability to groups that were relatively underrepresented, including younger adults and minoritized groups.”

R1.2 Authors have cited heterogeneity as an issue potentially taken care of by standardised protocols between cohorts. Could authors provide results of tests for heterogeneity and the potential consequences on data analysis? In addition, my understanding of table 1 is that three of the cohorts (ARIC, FHS and REGARDS) contribute to around 80% of the pooled sample. Are results different between each of the three large cohorts, and between those three large cohorts and the 11 other smaller cohorts? Is there an added value for the inclusion of the 11 smaller cohorts?

Response: To explore the issues of cohort heterogeneity and whether inclusion of a certain cohort or set of cohorts was particularly influential with respect to the overall results, we repeated the main analysis with adjustment for cohort (**Supplementary Table 7**) and, as suggested by the reviewer, with stratification by cohort (limiting to the 3 largest contributing cohorts [ARIC, FHS, REGARDS] and using a “leave-one-out” approach to exclude ARIC, FHS, or REGARDS; **Supplementary Figure 2**). Overall, the findings are quite consistent, and we therefore believe that cohort heterogeneity does not significantly bias our results. Nevertheless, the supplementary data will allow readers to make their own determinations. The following changes have been made to the text:

- *Methods*: “To explore whether inclusion of a certain cohort or set of cohorts was influential with respect to the overall results, we repeated the main analysis using a “leave-one-out” approach by separately removing participants in the three largest cohorts in this analysis

(ARIC, FHS, and REGARDS) from the pooled analysis sample. We also performed a sensitivity analysis in which we only included participants from ARIC, FHS, and REGARDS.”

- *Results:* “Findings were also consistent in models using different approaches to assess potential cohort heterogeneity, including models adjusting for cohort (**Supplementary Table 7**) and stratified models that were restricted to, or excluded, the three cohorts (ARIC, FHS, REGARDS) contributing the largest number of participants to the main analysis (**Supplementary Figure 2a-d**).
- *Discussion:* “Finally, cohort heterogeneity could influence our results, yet C4R ascertained COVID-related exposures using standardized measures, pre-pandemic characteristics were harmonized following established protocols, and cohort-adjusted and cohort-stratified sensitivity analyses yielded highly consistent results.”

R1.3 Authors suggest that variation in vaccine-induced antibody response may be clinically significant since lower humoral responses to SARS-CoV-2 vaccines are predictive of higher risk for breakthrough infections and severe COVID-19 -2 clinical outcome. Authors suggest that using the mean percent difference in anti-S1 MFI change provided a simple and clearer understanding of change in antibody levels. They may want to discuss, (beside statistical significance) the clinical and public health meaning of a specific, sometimes low, percent change in anti-S1 MFI.

Response: We appreciate the importance of discussing the potential public health and clinical implications of our findings while also acknowledging the current limitations of data available in our study and others regarding minimal clinically significant effect sizes for differences in anti-S1 levels. Unfortunately, we do not currently have the necessary data to assess the predictive significance of percent difference in anti-S1 MFI with respect to risk of severe COVID-19 and/or post-acute sequelae of SARS-CoV-2 infection (PASC). In this context, we have softened our language about clinical significance and provided cautious interpretations of our findings while acknowledging this limitation in the Discussion and highlighting this as a future research area.

- *Discussion:* “Second, almost all of our participants had antibody levels above the threshold for “reactivity” and it is unknown to what extent variation of antibody levels above that threshold is clinically meaningful. In future work, we intend to test if differences in anti-S1 IgG levels are associated with prospective risk of infection severity and PASC.”

R1.4 Due to missing values authors did an imputation. Could they provide details of the imputation method? In addition, in some occurrences (prior infection) the numbers of participants included in the respective analyses (imputed and not) are quite different and the validity of an imputation may be discussed.

Response: In response to this comment, we have added text in the Methods section that provides more details about the imputation method. We have also added a complete case analysis (**Supplementary Table 5**), a table showing missingness patterns for all relevant variables (**Supplementary Table 9**) and a table showing characteristics based on non-imputed data (**Supplementary Table 10**). The results were updated in many locations to note these changes. Overall, our findings between the imputed and complete case analyses were remarkably consistent in terms of strength and direction of associations for the risk factors explored. These

patterns suggest that it is unlikely for imputation to have created meaningful bias. The following text was added:

Results: “Results were similar in complete case analyses (**Supplementary Table 5**).”

Methods: “Multiple imputation by chained equations was applied to handle covariate missing data (**Supplementary Table 9**). Ten imputed datasets were created, and Rubin’s rule was applied to combine the estimates from the imputed datasets. Characteristics of complete cases were similar to the imputed (primary) dataset (**Supplementary Table 10**). Complete case analysis was performed as a sensitivity analysis.”

R1.5 In eTable for non imputed data, authors may want to show number and percent of missing information for each variable.

Response: We thank the reviewer for this suggestion and have now included a supplementary table showing the number of participants with missing covariate data (**Supplementary Table 9**).

R1.6 From Table 1, 9.1% of the participants only received one dose of vaccine. Do results differ when excluding them?

Response: In a new sensitivity analysis that excludes participants who received only one vaccine dose (**Supplementary Table 4**), the results are consistent with the main findings. In addition, Figure 1 includes only participants who had received a 2nd dose. We did not plot results for participants only receiving dose 1 because of the small number of participants who provided a dried blood spot after the first dose and before the 2nd dose.

- *Results:* “An analysis limited to participants with two doses of the BNT162b2 or mRNA-1273 vaccines yielded similar results compared to the main analysis (**Supplementary Table 4**).”

R1.7 From Table 2 authors suggest that, as time between first vaccine dose and DBS lengthened, no-reactivity is more prevalent among BNT162b2 than mRNA-1273 recipients. However, table 2 (which shows reactivity), suggests very small absolute differences in non-reactivity prevalence.

Response: We agree that the differences in non-reactivity prevalence are modest. While we have kept this information in the manuscript, we have moved these results to the supplement (**Supplementary Table 2**) and removed language contrasting the reactivity levels by vaccine type. We have also endeavored to acknowledge the current limitations of data available in our study and others regarding minimal clinically significant effect sizes for differences in anti-S1 levels, as discussed in **R1.3**. Relevant changes to the manuscript include the following:

- *Results:* “Among 6,491 participants sampled 30-270 days after their first dose of the vaccine, 94% showed antibody reactivity above the laboratory threshold used to confirm prior SARS-CoV-2 exposure (**Supplementary Table 2**).”
- *Discussion:* “First, information on neutralizing antibodies was not available, nor were other immunoglobulin isotypes (IgM, IgA) or measures of cellular immunity examined. Second, almost all of our participants had antibody levels above the threshold for “reactivity” and it is unknown to what extent variation of antibody levels above that threshold is clinically

meaningful. In future work, we intend to test if differences in anti-S1 IgG levels are associated with prospective risk of infection severity and PASC.”

R1.8 One of the major contributing factors to the level of anti-S1 IgG antibody levels is the occurrence of prior SARS-CoV-2 infection. Authors mention that documentation of prior infection was based on self-reporting and/or medical records review. Could authors describe what were the precise definitions of self-reported (clinical? Self-tests? Etc.) and medically recorded (RT-PCR confirmed?) prior infection? Computation of sensitivity and specificity of definitions (and related predictive values) may be relevant when adjusting for or stratifying analysis by prior infection. It may also affect results according to time since prior infection.

Response: As requested, we have clarified how SARS-CoV-2 infection was defined in C4R and we have added a sensitivity analysis in which we excluded cases of infection that were not confirmed by a positive SARS-CoV-2 test (ascertained either by self-report, medical record review, or serosurvey; **Supplementary Tables 6 and 8**). While computation of the sensitivity and specificity of these definitions is complex and outside the scope of the present work, the results in this sensitivity analysis are highly consistent with the main results. Also of note, we did not model time-since-infection since this was not the primary question of interest and some infected participants did not recall the date of infection. We have added an acknowledgement on how this may have affected our results for “breakthrough” infections. The relevant changes to the manuscript are as follows:

- *Results:* “Compared to uninfected participants, those with a self-reported history of infection prior to vaccination showed higher antibody levels... Anti-nucleocapsid antibody level, an objective marker of natural infection, was also associated with higher anti-S1 IgG. Infections occurring after vaccination (sometimes called “breakthrough” infections), but prior to the serosurvey... were associated with a lesser rate of antibody waning, as was higher anti-nucleocapsid antibody titer (**Supplementary Table 3**), which could be due to the fact that time since vaccination does not correspond to time since most recent SARS-CoV-2 antigen exposure in this group... Results were similar... in models excluding cases of self-reported infection that lacked confirmatory testing (**Supplementary Table 6**).”
- *Discussion:* “Infection history was defined primarily by self-report and could be subject to false-positive or false-negative classification, with uncertain influence on our estimates. We did not have data regarding health-related behaviors (e.g., masking and participation in large gatherings) that could correlate with probability of subclinical, unreported infection. Nonetheless, our models included anti-nucleocapsid antibody levels, which are an objective, sensitive biomarker of potential prior clinical or subclinical infection among vaccinated populations. Results in a sensitivity analysis limited to participants with a history of a positive COVID-19 test yielded similar results.”
- *Methods:* “C4R collected information on SARS-CoV-2 infection and vaccination status via two waves of questionnaires conducted from April 2020 through February 2023; the current report includes data from questionnaires collected through August 2022. Questionnaires were administered by telephone interview, electronic survey, in-person examination, and/or mailed pamphlet. Participants were asked about history of SARS-CoV-2 infection, SARS-CoV-2 testing, COVID-19 hospitalization, COVID-19 vaccination status, date of first vaccine administration, number of vaccines received, and vaccine manufacturer. Definite history of SARS-CoV-2 infection was classified based on self-report of a positive SARS-CoV-2 test,

adjudication of medical records for a COVID-19 hospitalization, or evidence of anti-nucleocapsid antibody reactivity on DBS; self-reported infections not meeting any of these criteria were classified as probable (**Supplementary Table 8**). Both definite and probable cases of infection were included in the main analyses, and a sensitivity analysis was performed that excluded probable cases.”

R1.9 In Figure 1 authors show anti-S1 MFI log transformed. As such, should not we expect a log Y axis instead of an arithmetic one? Same question for Figure 2.

Response: We have corrected the Y-axis label to read, “log-transformed anti-S1 MFI.”

R1.10 Authors suggest that mRNA-1273 (Moderna) vaccine induces a higher antibody response than the BNT162b2mRNA (Pfizer-BioNTech). In their discussion authors may want to acknowledge the different dosages between the two vaccines and discuss the consequences.

Response: We updated the Discussion to acknowledge the potential role of differences in dosages between the vaccine types.

- *Discussion:* “Compared to BNT162b2, the mRNA-1273 vaccine was associated with substantially higher antibody responses and slower antibody waning. Whether this may be due to differences in dosing between the two mRNA vaccines for the primary series (mRNA-1273: 100 mcg/dose; BNT162b2: 30 mg/dose), differences in the recommended dosing schedule, or other differences in vaccine formulations merits further investigation. Our findings may also have implications for dosing considerations relating to additional “booster” vaccines, since Moderna decreased the “booster” dose to 50 mcg.”

R1.11: Authors mention that anti-nucleocapsid antibody levels was associated with higher anti-S1 IgG. This is a difficult association to measure since anti-S1IgG measures both natural and vaccine induced antibodies.

Response: We included anti-N in our models because, in discussion with our infectious disease collaborators at the Wadsworth Center, it was thought that anti-N remains a biologically relevant surrogate marker of prior infection and may complement our more subjective assessment of prior COVID-19 infection in this vaccinated population (see response **R1.8**), which could have missed subclinical infections. We observed significant associations of both anti-N and a history of prior infection with higher anti-S1 levels, which suggests the potential role of infection in augmenting the vaccine response. Whereas there may have been more doubt if we relied only on self-assessments of infection.

R1.12 Overall, this study can be a major contribution to understanding duration of protection after vaccination while taking prior infection into account. My main suggestion to authors is to further discuss the representativeness of the non-random study sample and the potential consequences on the results. The understanding of the current results would ideally be complemented by an analysis of antibody levels and decline after infection in the unvaccinated C4R population.

Response: We appreciate the reviewer's enthusiasm for the potential contribution of our study to the literature. We acknowledge the under-sampling of certain groups like underrepresented minorities and younger adults may influence the generalizability of the findings. For a thorough description of our response to this point, please refer to response **R1.1b** above.

Reviewer #2 (R2)

R2.1 The current title uses the word "predictors" but I think this should be removed and replaced with, for example, association, as this is not a predictive study design, it is observational in nature.

Response: We agree and have modified the title of our manuscript to: “Demographic and Clinical Factors associated with SARS-CoV-2 Spike 1 Antibody Response Among Vaccinated US Adults: The C4R Study”

R2.2 It would be helpful if more background information on the cohorts and their representativeness as a general population sample was provided to aid readers of this paper as a standalone article, rather than having to refer to external literature.

Response: See our response below to **R2.4**. Also, we acknowledge that our findings may not be entirely generalizable to the U.S. population in the Discussion (see response to **R1.1b** above).

R2.3 My main concern with the analysis at present is regarding the lack of stratification of participants post first and second dose vaccine in analyses, particularly tables 1-3 and figures 1-5. There is good evidence of a quantitative difference in S level response between first and second dose so to combine these into one group for these descriptions of the anti-S1 levels seems inappropriate. This approach is inconsistent with most previous analyses including for e.g. Wei, J., Stoesser, N., Matthews, P.C. et al. Antibody responses to SARS-CoV-2 vaccines in 45,965 adults from the general population of the United Kingdom. *Nat Microbiol* 6, 1140–1149 (2021). <https://doi.org/10.1038/s41564-021-00947-3>

Response: In our initial approach, we considered the number of vaccine doses as a potential factor that influences anti-S1 IgG levels. We believe it may still be informative to readers to show that multiple doses of the initial COVID-19 mRNA vaccines induce higher anti-S1 IgG levels. However, in response to this comment, we have added a new sensitivity analysis that excludes participants who received only one vaccine dose (**Supplementary Table 4**), and the results are similar to the main findings. We added the following text to the manuscript:

- *Results:* “An analysis limited to participants with two doses of the BNT162b2 or mRNA-1273 vaccines yielded similar results compared to the main analysis (**Supplementary Table 4**).”

R2.4 Some additional information about the cohorts and how data were collected would be useful in terms of understanding risk of bias, including:

- How were cohorts originally sampled?
- How were diagnoses of COVID made?
- How was self-report data collected during the pandemic?
- How did the analyses account for across cohort heterogeneity?
- What is geographical spread of the cohorts?
- How were the cohorts followed up during covid and how did this compare to previous follow up methods?
- How was severe covid defined/classified?

- How much missing data was there across the cohorts and how was this dealt with?

Response: Many of these details are described in the C4R design paper (PMID: 35279711; reference 16). We have nonetheless added several details in the revision, including (1) cohort descriptions (**Supplementary methods: Cohort Descriptions**); (2) an expanded description of COVID-19 ascertainment (**Methods**), including our approach to classifying definite versus probable infections with a related sensitivity Analysis (**Methods, Supplementary Tables 6 and 8**); (3) additional analyses relevant to cohort heterogeneity, as described in response to **R1.2** (**Supplementary Table 7, Supplementary Figure 2**); and (4) analyses and discussion relevant to missing data in **R1.4** (**Supplementary Tables 5, 9, and 10**).

R2.5 Why was the group Novavax/AZ combined in table 1 and these individuals excluded elsewhere in the analysis?

Response: We thank the reviewer for drawing this to our attention. We have clarified that the “Other” group were participants who reported to have been vaccinated with (1) a vaccine that was not mRNA-1273 or Pfizer or (2) did not know the type of vaccine they received. This is the case for the main multivariable regression analyses and sensitivity analyses that are reported in the supplement. We have fixed this in all of the tables.

R2.6 Are the results in Table 3 univariate or multivariate with everything else in the table included in the model? If it's the later then this seems inappropriate.

Response: As suggested, we now present both unadjusted and multivariable-adjusted associations in **Table 2** and we have clarified the relevant methods and footnotes. Overall, similar associations are observed between the univariable and multivariable adjusted analyses. We did observe that parameter estimates for hypertension and the “Other” vaccine group became nonsignificant in multivariable models. Relevant changes include:

- *Results:* “Both unadjusted and multivariable-adjusted models consistently demonstrated associations between antibody levels and several participant characteristics (**Table 2**).”
- *Table 2:* “Correlates of anti-S1 IgG levels after COVID-19 vaccination in unadjusted and multivariable adjusted analyses...” Multivariable adjusted results arise from a regression model including all variables present in this table.”
- *Methods:* “Log-transformed anti-S1 IgG levels was regressed on candidate risk factors, time-since-vaccination (defined as days from first vaccination to DBS), and laboratory batch, using linear regression. Associations with candidate risk factors were tested with and without multivariable adjustment for other candidate risk factors under consideration.”

R2.7 Limitations should include a discussion on the higher proportion of females and older age profile of the combined cohort and issues regarding generalisability as a result of these biases.

Response: We have added discussion of potential selection biases and their impact on the findings in the discussion (see response to **R1.1b** for detailed comments). We have added the following text to the Discussion:

- *Discussion:* “Sixth, the meta-cohort was not directly representative of the US population and is subject to healthy volunteer bias, which likely biased our results towards the null and limits

generalizability to groups that were relatively underrepresented, including younger adults and minoritized groups.”

R2.8 Authors state that "UK Real-Time Assessment of Community Transmission-2 (REACT-2) study quantified antibody reactivity as a binary variable up to 20 weeks post-vaccination among a sample mostly receiving either BNT162b2 or ChAdOx1 vaccines." However they do not acknowledge that several other UK studies did use continuous measures including the UK ONS and Virus Watch studies.

Response: We thank the reviewer for drawing our attention to these publications and have incorporated them into our manuscript. We have added the following references in our revision:

1. Shrotri M, Fragaszy E, Nguyen V, Navaratnam AMD, Geismar C, Beale S, Kovar J, Byrne TE, Fong WLE, Patel P, Aryee A, Braithwaite I, Johnson AM, Rodger A, Hayward AC, Aldridge RW. Spike-antibody responses to COVID-19 vaccination by demographic and clinical factors in a prospective community cohort study. *Nat Commun.* 2022 Oct 2;13(1):5780.
2. Wei J, Stoesser N, Matthews PC, Ayoubkhani D, Studley R, Bell I, Bell JI, Newton JN, Farrar J, Diamond I, Rourke E, Howarth A, Marsden BD, Hoosdally S, Jones EY, Stuart DI, Crook DW, Peto TEA, Pouwels KB, Eyre DW, Walker AS; COVID-19 Infection Survey team. Antibody responses to SARS-CoV-2 vaccines in 45,965 adults from the general population of the United Kingdom. *Nat Microbiol.* 2021 Sep;6(9):1140-1149.

Also, we have made changes to our Discussion and remove statements about the continuous outcome of anti-S1 levels in the UK studies and thank the reviewer for directing us to these additional studies. Instead, we highlight a key difference of our study compared with others from the UK is that nearly 50% of our cohort reported to have taken the mRNA-1273 vaccine and we also highlight in our discussion the potential impact of different dosing.

- *Discussion:* “UK population-based studies were comprised mostly of individuals who received the BNT162b2 or ChAdOx1 vaccines; in contrast, almost all our participants received mRNA vaccines, and approximately half received the mRNA-1273 vaccine. Compared to BNT162b2, the mRNA-1273 vaccine was associated with substantially higher antibody responses and slower antibody waning. Whether this may be due to differences in dosing between the two mRNA vaccines for the primary series (mRNA-1273: 100 mcg/dose; BNT162b2: 30 mg/dose), differences in the recommended dosing schedule, or other differences in vaccine formulations merits further investigation. Our findings may also have implications for dosing considerations relating to additional “booster” vaccines, since Moderna decreased the “booster” dose to 50 mcg.”

R2.9a "Consistent with prior literature, we observed that anti-S1 levels peaked at around 60 days after vaccination with a subsequent decline." This does not seem to be consistent with existing literature from the UK where both ONS and Virus Watch show a much earlier peak post first and second dose. The authors should re-examine this finding after splitting participant data into first and second dose responses.

Response: We have carefully reviewed the referenced studies. Our interpretation of results from the two UK studies vs. C4R suggest that antibody peaks in C4R occurred sooner than in Virus Watch but later than in ONS, and that waning occurred more rapidly in C4R. We have updated the discussion to describe these differences and discuss potential explanations.

- *Discussion:* “Relative to UK data, our C4R data show peak antibody levels occurring at approximately 60 days and antibody waning was evident by 90 days. This suggests that the US peak was sooner than in the Virus Watch and REACT-2 cohorts, in which the peak occurred closer to 90 days after the first dose, but later than in the ONS cohort, in which the peak occurred approximately 28 days after the first dose. Nonetheless, the timing of the peak antibody response in C4R was similar to that observed in a longitudinal US-based study of BNT162b2 recipients. In our US-based cohort, waning occurred more rapidly relative to the UK population, in which anti-spike IgG levels showed no obvious sign of waning by 90 days. Of note, longer intervals between the first and second dose implemented in the UK (8-12 weeks) versus in the US (3-4 weeks) have been hypothesized to provoke a more durable immune response; this issue deserves more careful attention in future research studies. The faster waning observed in our sample may also be attributable to the fact that C4R participants are older and have more comorbidities linked to immunosenescence. Another important distinction between data collected in C4R vs. UK cohorts relates to the timing of serological assessments: published UK studies occurred primarily in 2021, while C4R results span from mid-2021 to mid-2022. Finally, UK studies collected multiple within-person longitudinal samples, whereas C4R only has one serological assessment for each person, hence our findings could be biased by confounding by unmeasured factors related to both timing of vaccination and antibody response.”

R2.9b "...The authors should re-examine this finding after splitting participant data into first and second dose responses.”

Response: Although the group with only one vaccine dose is underpowered, we have limited Figure 1 to the group receiving two doses and repeated linear regressions in a sensitivity analysis limited to the two-dose group, yielding similar findings compared to the main analyses.

- Results: “Antibody levels were highest at 60 days following first vaccine dose, after which antibody levels waned, on average, at a rate of 23% per month (**Figure 1**).”
- Results: “Self-report of a single dose of vaccine (N=499, 7.3%) was associated with a lower anti-S1 IgG level compared to two doses (**Table 2**). An analysis limited to participants with two doses of the BNT162b2 or mRNA-1273 vaccines yielded similar results compared to the main analysis (**Supplementary Table 4**).”
- Methods: “Analyses were also repeated with exclusion of participants reporting only one dose of vaccine.”

R2.10 "Strategies to optimize COVID-19 vaccine responsiveness at the individual level, and deployment at the population level, warrant further research and development. " Can the authors be more specific and draw specifically from their results in terms of which issues require further research and development.

Response: We have added a more specific recommendation about groups to target in the revised Discussion.

- *Discussion:* “Taken together, our results suggest that additional (or higher) vaccine doses should be particularly encouraged for individuals at risk of lower anti-S1 responses to vaccination, including older individuals and those with obesity, diabetes, and/or COPD. Future research will be important to determine if more frequent vaccination or higher dosing in these vulnerable groups are effective in preventing severe COVID-19”

Reviewer #3 (R3)

R3.1 Line 265: I'm not sure what do you mean by "As the time between first vaccine dose and DBS lengthened". Non-reactivity was more prevalent among those who received BNT162b2 vaccine and those 80 years or older across all groups defined by similar time between first vaccine dose and DBS.

Response: As suggested, we have removed this sentence and have moved the tabular summary of reactivity by time since first dose and vaccine type to **Supplementary Table 2**. Please see responses **R1.3** and **R1.7** for discussion of how we have framed our results on reactivity versus antibody response.

R3.2 Line 276: "Asian race was associated with greater antibody response". It should be compared to non-Hispanic white.

Response: We have edited the text to include the correct reference groups for all comparisons. For example:

- *Discussion:* "...no consistent associations with race or ethnicity were observed, with the exception of higher antibody levels in participants reporting Asian versus non-Hispanic White race; this subgroup was small, hence the finding should be interpreted with caution, but it is somewhat consistent with a prior study on higher antibody levels in Asian participants."

R3.3 Line 279: What is "fully adjusted models"? Is the model used for health behaviors and comorbidities different from the rest of the variables. From your description of statistical analysis, it seems like there is only one model for antibody response ("Log-transformed anti-S1 MFI was regressed on candidate risk factors using multivariable linear regression models, adjusted for time-since-vaccination and laboratory batch.")

Response: We have clarified our modeling approach in the methods as well as in the footnotes for our tables. Revised Table 2 includes both unadjusted and multivariable adjusted results. Our 'fully adjusted' multivariable results are derived from models that simultaneously adjust for all variables presented in Table 2.

R3.4 "Line 279: "Lower antibody responses were observed in former smokers (6.7%, 95% CI,0.9-12.2) and current smokers (15.2%, 95% CI, 5.9-23.7). " Again, please provide the "reference group in the text."

Response: We have edited to include the correct reference groups for all comparisons.

R3.5 Line 284: "SARS-CoV-2 infection was strongly associated with higher anti-S1 IgG antibody response." Should it be prior COVID-19 disease instead of SARS-CoV-2 infection? If "hospitalized" means "severe" and "non-hospitalized" means "not severe" in Table 3, then please be consistent in the text and the table."

Response: Since we have included asymptomatic infections, we have edited the text to refer to SARS-CoV-2 infections, which are sub-classified as hospitalized or non-hospitalized. The term “severe” has been replaced with “hospitalized” throughout the manuscript.

- *Methods:* “Infections were dichotomized according to whether they were, or were not, associated with hospitalization.”

R3.6 Line 302: What do you mean by “Based on the largest effect estimates from multivariable regression analysis”? Did you include only variables with the largest effect estimates?”

Response: In response to this comment, and for completeness, we now present plots of antibody response over time for all clinical factors that were associated with differential antibody response in the multivariable-adjusted model: age, sex, BMI, smoking, diabetes, and COPD status (**Figure 2**), vaccine type (**Figure 3**) and prior infection status (**Figure 4**).

R3.7 "What is the percentage of missing covariables in your model?"

Response: We have now added this information to the manuscript. Please see **Supplementary Table 9** as well as our detailed response to **R1.4** on the issue of missing data.

Reviewers' Comments:

Reviewer #1:

Remarks to the Author:

This study is a relevant contribution to the identification of personal characteristics associated with lower anti-S1 antibody response and decrease over time after COVID-19 vaccination. Such studies, in complement to post marketing vaccine effectiveness studies, help to understand duration of protection and to identify subpopulation who could potentially benefit from adapted vaccination schedules. The study is based on the pooled analysis of 14 cohorts that is described as a collaborative cohort of cohorts in the US population.

Following the first review, authors have addressed the 11 points that I had raised about either clarification, specific methodological issues, potential biases and representativeness of the proposed results.

Each of the 11 responses is based on a clear understanding of the questions I raised, documented and quantified responses as needed. This is followed by important modifications of the manuscript (methods, results and discussion). The conclusions have been softened to take into account potential biases and uncertainty.

Specific responses addressed the following issues:

R1.1A. Description of characteristics of higher or lower level of anti-S1. Authors have mostly standardised the way they describe the outcome (lower or higher level). The few remaining exceptions are clear and do not lead to misunderstanding.

R1.1B. Absence of randomisation for participants selection.

Potential lack of representativeness is answered using a table showing baseline characteristics of C4R participants not included in the analysis (Supplementary Table 1).

R1.1C. Healthy cohort effect. Authors have discussed this issue in point 6 of their discussion chapter.

R1.2. Heterogeneity between cohorts contributing to C4R. The point has been clarified and is supported by Supplementary table 7, supplementary figures 2a-d and several sensitivity analyses.

R1.3. Predictive public health and clinical significance of percent difference in antibody levels. Authors acknowledged the points and mention that necessary data are not currently available. This was discussed in the second point of discussion. The conclusion language has been softened accordingly.

R1.4 Imputation method. Authors have now described the imputation method used. The issue is supported by supplementary tables 5, 9 and 10.

R1.5. Number and percent of missing values. Number and percent per characteristics are now included in supplementary Table 9.

R1.6. Are results different when excluding participants receiving only one dose of vaccine? A sensitivity analysis has been done and yield similar results.

R1.7. Modest difference in non-reactivity prevalence between BNT162B2 and mRNA-1273. This was moved to supplementary material and language contrasting reactivity levels by vaccine was softened (supplementary table 2, first and second point discussion).

R1.8. Definitions of "prior infection". SARS-CoV-2 infection definition was clarified. Relevant aspects were added in Results, Discussion and Methods. The question raised about Se and Sp of definitions and consequences could not be addressed in the current study but remains an issue for any study based on large electronic data sets.

R1.9. Figure 1. Authors have modified the figure and used a Log Y axis instead of the arithmetic one originally proposed.

R1.10. Higher antibody response of mRNA-1273 (Moderna) vaccine may be partly due to higher dosage? Authors have appropriately discussed the issue in the discussion section.

R1.11. Anti-S1 and anti-N. Authors provide useful explanation of their choice.

R1.12. "Overall, this study can be a major contribution to understanding duration of protection after vaccination while taking prior infection into account. My main suggestion to authors is to further discuss the representativeness of the non-random study sample and the potential consequences on the results. The understanding of the current results would ideally be complemented by an analysis of antibody levels and decline after infection in the unvaccinated C4R population."

The above comment, made after the first review, is strengthened thanks to the re-analyses,

explanations and modifications brought into the wording (soften) and supplementary material provided. With these changes the manuscript brings a useful contribution to understanding the association between potential personal characteristics, prior infection with SARS-CoV-2, vaccinations and subsequent levels of immunity.

As such, taking only into the comments and suggestions I had made in the first review and the related authors' responses and modifications, I consider that the current manuscript is improved and suitable for publication.

Reviewer #2:

Remarks to the Author:

My main concern with the analysis remains: the authors present the results without a lack of stratification of participants post first and second dose vaccine in analyses (or alternatively the exclusion of all participants with just one dose from all analyses if the sample size isn't large enough for stratification). The included additional supplementary table 4 does not remove my concerns as I think the presentation of figures 1-5 could be particularly affected by this issue. I still remained concerned about the Tables too.

There is substantial evidence of a quantitative difference in S level response between first and second dose. To combine these into one group for these descriptions of the anti-S1 levels seems inappropriate, is inconsistent with many previous analyses, and as a result substantially impacts the interpretability of the findings.

Reviewer #3:

Remarks to the Author:

I thank the authors for addressing my comments.

Reviewer #2 (R2)

R2.2 My main concern with the analysis remains: the authors present the results without a lack of stratification of participants post first and second dose vaccine in analyses (or alternatively the exclusion of all participants with just one dose from all analyses if the sample size isn't large enough for stratification). The included additional supplementary table 4 does not remove my concerns as I think the presentation of figures 1-5 could be particularly affected by this issue. I still remained concerned about the Tables too.

There is substantial evidence of a quantitative difference in S level response between first and second dose. To combine these into one group for these descriptions of the anti-S1 levels seems inappropriate, is inconsistent with many previous analyses, and as a result substantially impacts the interpretability of the findings.

Response: As suggested, we now exclude all participants who received only one dose of the COVID-19 vaccine, since the sample size among those with only 1 dose was too small to include as a subgroup. Also, to minimize heterogeneity and thereby strengthen interpretability, we have restricted our analysis to participants who received the mRNA COVID-19 vaccines (BNT16b2 or mRNA-1273). All our analyses have been updated and we have removed the two-dose restricted analysis from our first revision (Revision #1-Supplementary Table 4). We also recategorized age into three groups (<65 years, 65-79 years, \geq 80 years), which we believe will make our results more interpretable for the readers as the <50 years age subgroup was much smaller. Overall, our analysis remains very similar to our original manuscript and our first revision. Below is a list of our main findings in our updated analysis:

- Older age, male sex, smoking history, diabetes, body mass index, COPD, and longer time between first vaccine dose and serosurvey collection remained associated with lower levels of anti-S1 IgG in univariable and multivariable analysis (Table 2).
- Anti-N, prior COVID-19 infection history before vaccination, and mRNA-1273 vaccine type (compared with BNT162b2 mRNA) were each associated with higher anti-S1 levels (Table 2).
- The generalized additive model plots are similar to our figures from Revision #1 after excluding participants with only one dose of the vaccine (Figures 1-4).
- Most of the significant findings from Table 2 remained significant in our sensitivity analyses (Table S5-excluding non-definite cases, Table S6-adjustment for cohort, Figure S2-exclusion of largest cohorts).
- Compared with non-Hispanic White subgroup, self-reported AIAN is now associated with lower anti-S1 IgG levels in the univariable and multivariable analysis (Table 2).